# *Theileria* parasites sequester host eIF5A to escape elimination by host-mediated autophagy

Marie Villares[1], Nelly Lourenço[1], Ivan Ktorza[1], Jérémy Berthelet[1], Aristeidis Panagiotou[1], Aurélie Richard[1], Angélique Amo[1], Yulianna Koziy[1], Souhila Medjkane[1], Sergio Valente [2], Rossella Fioravanti[2], Catherine Pioche-Durieu [3], Laurent Lignière[3], Guillaume Chevreux[3], Antonello Mai [2,4] & Jonathan B. Weitzman [1] ✉

Intracellular pathogens develop elaborate mechanisms to survive within the hostile environments of host cells. *Theileria* parasites infect bovine leukocytes and cause devastating diseases in cattle in developing countries. *Theileria* spp. have evolved sophisticated strategies to hijack host leukocytes, inducing proliferative and invasive phenotypes characteristic of cell transformation. Intracellular *Theileria* parasites secrete proteins into the host cell and recruit host proteins to induce oncogenic signaling for parasite survival. It is unknown how *Theileria* parasites evade host cell defense mechanisms, such as autophagy, to survive within host cells. Here, we show that *Theileria annulata* parasites sequester the host eIF5A protein to their surface to escape elimination by autophagic processes. We identified a small-molecule compound that reduces parasite load by inducing autophagic flux in host leukocytes, thereby uncoupling *Theileria* parasite survival from host cell survival. We took a chemical genetics approach to show that this compound induced host autophagy mechanisms and the formation of autophagic structures via AMPK activation and the release of the host protein eIF5A which is sequestered at the parasite surface. The sequestration of host eIF5A to the parasite surface offers a strategy to escape elimination by autophagic mechanisms. These results show how intracellular pathogens can avoid host defense mechanisms and identify a new anti-*Theileria* drug that induces autophagy to target parasite removal.

The environment within host cells presents a formidable challenge for the survival of obligate, intracellular pathogens and invading microbes must develop sophisticated strategies to survive in these hostile environments[1]. Upon microbial entry, host cells mount a series of defensive mechanisms, including the innate and acquired immune machineries to counter the invaders[2]. Unicellular, eukaryotic parasites of the Apicomplexa phylum have evolved to become master manipulators of their host cells, exploiting cellular signaling pathways to hijack host gene expression in order to survive. The Apicomplexan parasites include major pathogens such as *Plasmodium spp*. that cause malaria in humans and *Toxoplasma* that infects over 1 billion people worldwide. *Theileria spp*. parasites infect bovine leukocytes and cause

[1]Université Paris Cité, CNRS, UMR7126 Epigenetics and Cell Fate, Paris 75013, France. [2]Department of Drug Chemistry & Technologies, Sapienza University of Rome, Rome 00185, Italy. [3]Université Paris Cité, CNRS, UMR 7592 Institut Jacques Monod, Paris 75013, France. [4]Pasteur Institute, Cenci-Bolognetti Foundation, Sapienza University of Rome, Rome 00185, Italy. ✉e-mail: jonnyw4@gmail.com

devastating diseases in cattle in developing countries[3]. *Theileria* spp. have evolved multiple strategies to hijack host bovine leukocytes, inducing proliferative and invasive phenotypes characteristic of cancer cell transformation[1,3,4]. Unlike most other Apicomplexan parasites, *Theileria* parasites reside in the cytoplasm of their host cells and notably lack a parasitophorous vacuole membrane (PVM). The absence of the PVM makes them particularly vulnerable to host defense mechanisms and offers an opportunity for direct contact with the host cellular proteins and cell machinery[5].

This direct contact between *Theileria* parasites and the host cell cytoplasm may be a driver for unique mechanisms of hijacking host cell pathways[4–6]. On the one hand, there is evidence that intracellular *Theileria* parasites secrete parasite proteins into the host cell cytoplasm that can stimulate oncogenic signaling pathways and some may even function within the host cell nucleus to drive transcriptional regulation[5–7]. For example, we identified a parasite-encoded prolyl isomerase TaPin1 that is secreted by *Theileria annulata* parasites and leads to the activation of host transcription factors, such as AP-1 and HIF1α, by hijacking host ubiquitin ligase machineries[8–10]. TaPin1-activated signals are implicated in the proliferative and metabolic pathways that contribute to the transformation of host leukocytes[11,12]. Notably, TaPin1 is at least one of the targets of the anti-parasite drug Buparvaquone and the gene encoding TaPin1 was found mutated in drug-resistant parasites in Tunisia and Sudan[8,13]. On the other hand, an alternative proposed strategy to hijack the host cell signaling machineries is sequestration of host proteins to the intracellular surface of the parasite macroschizont structure within the host cytoplasm[14]. Striking examples are the sequestration of the host IkappaB kinase (IKK) to mediate inflammatory NFκB signaling and the recruitment of host microtubule associated factors[14–16]. Other mechanism may exist, such as the recent description of nuclear pore-like complexes recruited close to the schizont surface[5]. Thus, several studies have begun to identify secretion and sequestration strategies to hijack host cell pathways and to identify the molecular players that lead to specific host cell outcomes.

Despite these advances, there are many aspects of the host defense against *Theileria* infection that remain unexplored. For example, it is well-established that autophagy is a powerful host defense mechanism to fight virulent intracellular pathogens[17,18]. Mechanisms to combat autophagy have been identified in *Plasmodium* parasites to survive within host hepatocytes and these involve sequestration of host LC3 (microtubule-associated protein 1 light chain 3) to the PVM surface[19]. Autophagy likely represents a first-line of attack against intracellular *Theileria* parasites, but the mechanisms involved remain unclear[20]. We discovered fortuitously a role for autophagy in eliminating intracellular *Theileria* parasites. Here, we describe a novel mechanism by which the intracellular *Theileria* parasite sequesters the host protein eIF5A to the macroschizont surface and prevents autophagic flux and parasite degradation. Notably, during a screen for new anti-theilericidal drugs, we identified a compound that induces an AMPK-dependent pathway leading to the release of eIF5A from the parasite surface and subsequent parasite elimination. Interestingly, this drug is the first to uncouple host cell survival from parasite survival and offers an alternative to the widely-used Buparvaquone drug. This study adds eIF5A to the growing list of host proteins that are sequestered to the parasite surface and offers novel insights into how sequestration could prevent, rather than activate, key cellular pathways leading to parasite survival and escape from host cell defenses.

## Results

### *Theileria* parasite survival can be uncoupled from host leukocyte survival

In order to search for new drugs against *Theileria* parasites which might overcome emerging resistance to the Buparvaquone reference drug, we screened a library of 150 compounds (that includes chromatin modifying compounds, histone deacetylase and demethylase inhibitors) first developed as anti-cancer drugs and then found active as antimicrobial agents[21–23]. We used a recently developed microscopy-based screening strategy[24] that measured host cell survival, the intracellular parasite load, schizont structures and the levels of a novel parasite histone marker, H3K18me, associated with the schizont stage[25]. Examples of the microscopy screen images are shown in Supplementary Fig. 1a. Previous studies suggested that the survival of the bovine leukocytes transformed by *Theileria* parasites is completely dependent on the survival of intracellular parasites[12]. However, our screen revealed several compounds that reduced the number of parasite nuclei with little impact on host macrophage survival (at the initial 10 μM screening concentration) [Fig. 1a]. The best three 'hit' compounds (MC2645, MC2646 and MC3205) represent the same chemical family and are all derivatives of the same parental compound[26]. We focused our subsequent validation and characterization on one of these, referred to as MC2646 [Supplementary Fig. 1a], as it gave the most reproducible results in validation screening. We validated the screen results, testing the MC2646 compound (1 μM for 48 h) on infected macrophages (Tac12 cells), or B lymphocytes infected with either *T. annulata* (TBL3 cells) or with the *T. parva* species (TpMD409 cells). The MC2646 compound effectively reduced the number of parasite nuclei, irrespective of the species or the host cell-type, and was equivalent to treatment with the Buparvaquone reference drug [Fig. 1b]. We investigated the effects of the MC2646 drug on the host cell cycle. Flow cytometry analysis showed that the MC2646 had no significant impact on host cells, whereas the Buparvaquone drug induced growth arrest in Tac12 macrophages and apoptosis (sub G1 population) in TBL3 lymphocytes (as previously reported[12]) [Fig. 1c and Supplementary Fig. 1b]. Neither drug affected parental BL3 cells. Cell proliferation assays confirmed that MC2646 did not affect host cell growth even after several days of treatment, in contrast to Buparvaquone [Fig. 1d]. To our knowledge, this is the first time that pharmacological intervention could uncouple parasite and host survival, raising new questions about the interdependence of parasite and host cells for the *Theileria*-induced transformation. To study the effect of MC2646 treatment on the parasite life cycle, we induced merogony by incubating infected Tac12 macrophages at 41 °C for 8–10 days. Merogony was measured by parasite load and the upregulation of the *TamR1* marker gene. MC2646 treatment completely blocked differentiation of remaining parasites and TamR1 expression [Fig. 1e]. As *Theileria* parasites induce a transformed phenotype in host leukocytes[3,5], we tested the impact of MC2646 on host cell colony formation in soft agar medium. Infected TBL3 cells formed numerous colonies, in contrast to uninfected parental BL3 cells [Fig. 1f]. Treatment with MC2646 blocked colony formation (despite the presence of live cells), as did the Buparvaquone drug [Fig. 1f]. Thus, MC2646 appeared to reduce host cell transformation, but not host survival. We performed RNA-Seq analysis to study the impact on the bovine transcriptome. A large number of induced and suppressed bovine genes are associated with *Theileria* infection[27] [Fig. 1g inner circle]. Transcriptome analysis showed that many of these events are reversed by treatment with Buparvaquone, but MC2646 treatment had a relatively modest effect on host gene expression [Fig. 1g]. Further analysis of gene expression patterns revealed a difference in the response of the KEGG Pathway in Cancer genes; again, Buparvaquone treatment had a greater effect than MC2646, notably for parasite-induced genes linked to transformation, such as *mmp9* [Fig. 1h]. Thus, we identified MC2646 as a compound [Supplementary Fig. 1c] that reduces parasite survival without affecting host cell survival.

### Intracellular *Theileria* parasites suppress host autophagy to survive in host cells

To investigate further the effect of MC2646 on host cells, we performed electron microscopy (EM) analysis of infected and uninfected

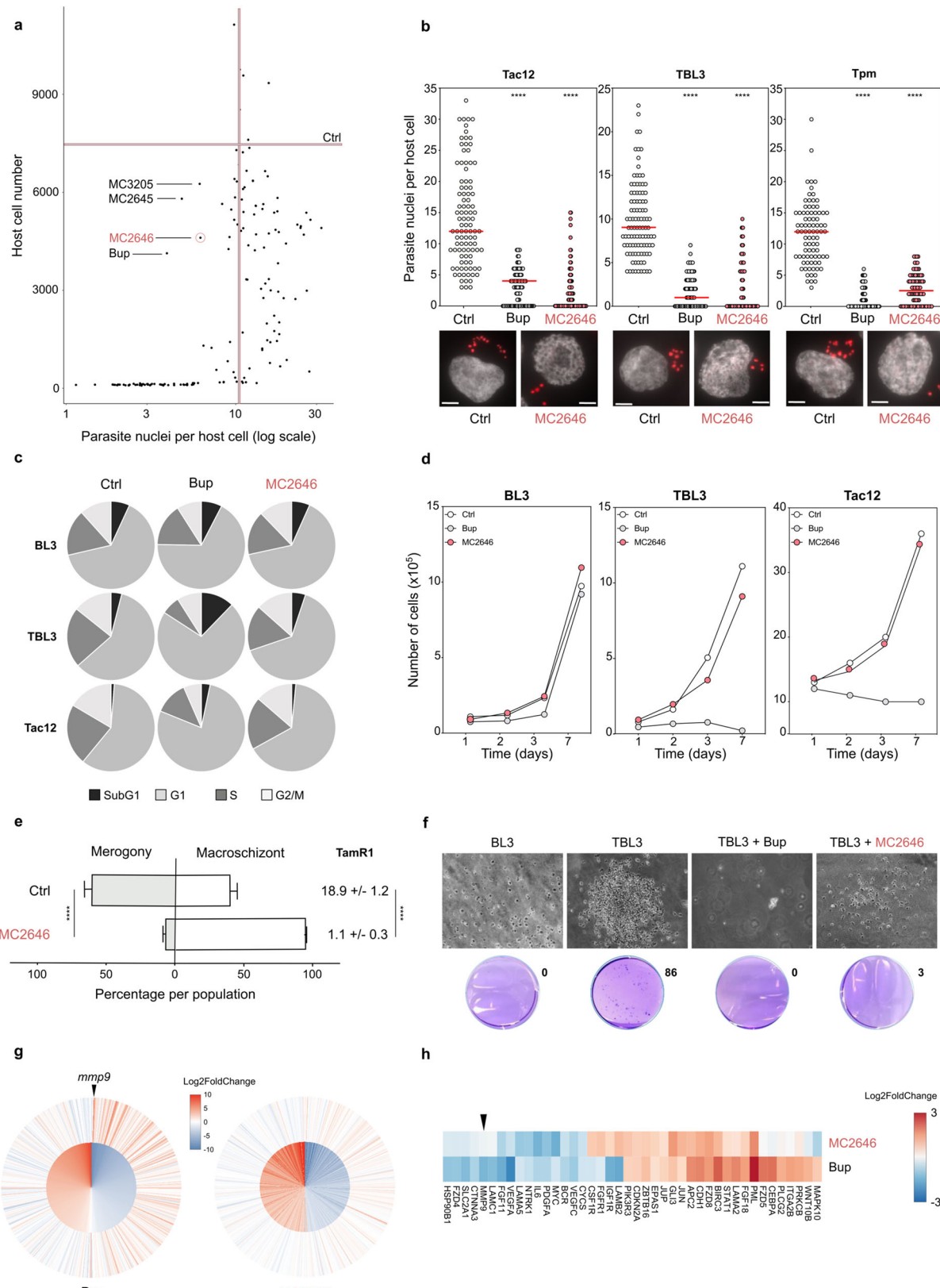

cells with drug treatment. The EM images revealed a marked difference between cells infected with parasites and uninfected cells and the effect of drug treatment; infected TBL3 cells showed a notable lack of structures linked to autophagosomes or autolysosomes, which were clearly visible in uninfected BL3 parental cells; notably, these structures re-appeared in TBL3 cells upon MC2646 treatment [Fig. 2a]. We

also observed the previously reported host annulate lamellae membrane structures around the parasite[28]. We hypothesized that the intracellular parasite may suppress host cell autophagy, which can be rescued by drug treatment. Gene Signature Enrichment Analysis (GSEA) of our RNA-Seq data confirmed a link with autophagic processes, showing that MC2646 treatment impacted autophagy-related

**Fig. 1 | A compound screen identifies MC2646 which uncouples parasite survival from host survival. a** Screening of a library of 150 compounds (10 μM for 48 h) monitoring host cell number and parasite nuclei number. Graphical representation of parasite and host survival per well. The number of parasite nuclei and host cells in DMSO controls are indicated by vertical and horizontal lines, respectively. Treatment with Buparvaquone (Bup) or three others compounds (MC2645, MC2646, MC3205) reduced parasite survival. **b** Treatment of *Theileria*-infected cell lines (Tac12 infected macrophages, TBL3 infected B lymphocytes, and Tpm *T. parva* infected lymphocytes) with MC2646 or Buparvaquone (Bup) for 48 h, compared to untreated controls (Ctrl). Quantification and representative images showing parasite nuclei marked with parasite-specific histone H3K18me1 and DAPI. At least 50 host cells were counted per condition. One way-Anova, Dunnett's multiple comparison test, ****$p < 0.0001$. Leica microscope x100 magnification; the scale bar corresponds to 5 μm. **c** Flow cytometry analysis of the cell cycle in uninfected B lymphocytes (BL3), infected lymphocytes (TBL3) or macrophages (Tac12) treated with Buparvaquone (Bup) or MC2646, compared to untreated controls (Ctrl). The percentage of G1, S phase, G2/M and sub-G1 populations are shown for each condition. Representative, flow cytometry data are shown in Supplementary Fig. 1b. **d** Growth curves of infected (TBL3 or Tac12) cells or uninfected (BL3) cells treated with Buparvaquone (Bup) or MC2646. Host cell viability was measured each day and compared to untreated controls (Ctrl). **e** MC2646 blocks parasite differentiation to merogony. Tac12 infected macrophages were incubated at 41 °C for 10 days (Ctrl) or treated with MC2646. Merogony was monitored visually and by the merogony marker gene TamR1 q-PCR analysis. Two-way Anova, Sidak's multiple comparisons test, ****$p < 0.0001$. **f** MC2646 blocks the transformed phenotype. Colony growth in soft-agar assay was monitored for infected (TBL3) or uninfected (BL3) cells with or without (Ctrl) addition of MC2646 or Buparvaquone. **g** MC2646 treatment has a modest effect on the parasite-induced bovine transcriptome. Differential expression analysis of the RNA-Seq datasets; the inner circle shows the top 500 bovine genes most significantly (*p*-val adjusted) impacted by infection (TBL3 vs BL3) and the outer circle shows the effect of drug treatment for each gene. All genes are ordered by descending Log2 fold change (TBL3 vs BL3) for the 1st dataset. The color scale represents Log2Fc, as shown in the legend. The graph was plotted using ggplot2 (v3.3.6) in R v4.1.1. Differential statistical analysis with Weighted Kolmogorov-Smirnov Test. **h** Differential impact of MC2646 or Buparvaquone on expression of genes in the KEGG Pathways in Cancer geneset. Color scale represents Log2Fc. Genes not detected across datasets, as well as genes with sum absolute Log2Fc < 1.5 were omitted. The genes were then clustered with the Ward.D method, and plotted using pheatmap (v1.0.12). In all experiments (apart from Fig. 1a) cells were incubated with 50 ng/ml Buparvaquone or 1 μM MC2646. All results are representative of 3 independent experiments. Statistical analysis Dunnett's multiple comparison test. ****$p < 0.0001$.

gene sets in infected cells (and to an even greater degree in uninfected cells, where autophagy was not suppressed), in contrast to Buparvaquone treatment [Fig. 2b]. To examine the autophagic state in our cells, we compared the LC3-I vs LC3-II levels. Western blot analysis revealed that the LC3-II form was lower in infected TBL3 cells (demonstrated by the altered LC3-II:LC3-I ratio) and the LC3-II:LC3-I ratio was increased almost two-fold by MC2646 treatment in both cell lines [Fig. 2c], consistent with a suppression of host cell autophagy by *Theileria* parasites. To test whether MC2646 reactivates autophagy, we analyzed the drug impact on LC3-puncta formation in the host cytoplasm; immunofluorescence analysis of the LC3B profile after methanol fixation showed small dots in the cytoplasm, corresponding to autophagosomes, of TBL3 cells, but only following MC2646 drug treatment [Fig. 2d]. These results support the intriguing possibility that *Theileria* parasites survive by escaping host cell autophagy[20]. To test this hypothesis, we used another classical method to induce host cell autophagy; we starved the TBL3 infected cells in low-nutrient media and measured parasite load. Incubation in EBSS media led to a significant decrease in parasite survival (as measured by parasite nuclei numbers in infected lymphocytes or macrophages) and this could be blocked by treatment with Bafilomycin A1 (BafA1), a known inhibitor of the latter stages of autophagy, affecting fusion between autophagosomes and lysosomes [Fig. 2e, Supplementary Fig. 2a]. The observation that BafA1 blocks the EBSS-induced parasite loss more than it does MC2646-induced parasite loss [Fig. 2e], suggests that MC2646 may also induce other pathways. However, one confounding factor is the timing of the different drug incubations. We treated cells for 24 h with the MC2646, before adding BafA1 for the last 2–3 h. This means that the effects of MC2646 on parasite survival may already be advanced before the addition of BafA1. Our results suggest that the suppression of host cell autophagy by intracellular *Theileria* parasites is necessary for parasite survival and the MC2646 compound rescues autophagic flux to eliminate parasites.

### The anti-parasite MC2646 compound activates the AMPK/TFEB pathway, autophagic flux and host autophagy processes to eliminate parasites

To characterize the host pathways activated by MC2646 treatment we tested a range of autophagy activators or inhibitors on parasite survival [Fig. 3a]. Rapamycin and Torin1 are well-characterized mTOR inhibitors which activate autophagy[29], whereas chloroquine (CQ) and BafA1 are inhibitors of autophagic flux; the former inhibits lysosomal acidification, the latter prevents fusion between autophagosome and lysosomes[30]. The mTOR inhibitors Rapamycin and Torin1 had no significant effect on the number of parasite nuclei per cell in parasitized TBL3 cells [Fig. 3a], suggesting that this autophagic pathway is inactive in TBL3 cells. Indeed, Rapamycin and Torin1 induced autophagic flux in control BL3 cells (as demonstrated by LC3 puncta formation in the presence of BafA1) [Supplementary Fig. 2b], but had no effect in parasitized TBL3 cells [Supplementary Fig. 2c-d]. Inhibition of the mTOR pathway is not sufficient to explain MC2646-induced effects on autophagy and parasite survival. The compound 991 activates the AMPK pathway, a regulator of autophagy[31]. Notably, we found that 991 treatment significantly reduced parasite nuclei number per host cell in infected cells (although less than Buparvaquone or MC2646) [Fig. 3a and Supplementary Fig. 2e-f]. The effects of 991 or MC2646 on parasite survival were rescued by treatment with the AMPK/ULK inhibitor, SBI-0206965 [Supplementary Fig. 2f]. We observed an increase (two-and-half-fold) in the amount of phosphorylated AMPK forms in TBL3 cells treated with either compound 991 or MC2646 (4 μM for 24 h) [Fig. 3b and Supplementary Fig. 2e]. Activation of the AMPK pathway induces, amongst other things, release of the transcription factor TFEB for nuclear translocation[32]. As MC2646 treatment appears to reinitiate autophagy and activate the AMPK pathway, we tested whether MC2646 treatment affects TFEB translocation. Immunofluorescence analysis showed that there was an increase in the nuclear:cytoplasmic TFEB localization upon MC2646 treatment [Fig. 3c]. We conclude that treatment with the MC2646 compound leads to activation of the AMPK pathway and the translocation of TFEB to the nucleus to participate in autophagy processes that reduce parasite survival.

To confirm that *Theileria* parasites suppress host autophagy and that the MC2646 drug rescues it to eliminate parasites, we studied autophagosome formation and autophagic flux in uninfected and infected cells. The LC3B protein exists in two forms, LC3-I and LC3-II; the latter is the gold-standard marker of autophagosomes[33]. We treated infected TBL3 cells with MC2646 or Buparvaquone, combined with BafA1 (50 nM for 3 h), followed by immunofluorescence of LC3B with methanol fixation or Western blot analysis. We saw few autophagosomes in untreated TBL3 cells, but MC2646 incubation led to a marked increase in LC3-puncta (and LC3-p62 autophagosome puncta) in TBL3 cells, which was even more pronounced upon the addition of BafA1 [Fig. 3d, Supplementary Fig. 3]. Similar results were observed by Western blot analysis of LC3B isoforms [Fig. 3e]. In contrast, Buparvaquone treatment did not increase LC3 puncta [Fig. 3d, and Supplementary Fig. 3c] or significantly impact the LC3-II:LC3-I ration [Fig. 3e] in the presence of BafA1. The BafA1 experiments demonstrate that the

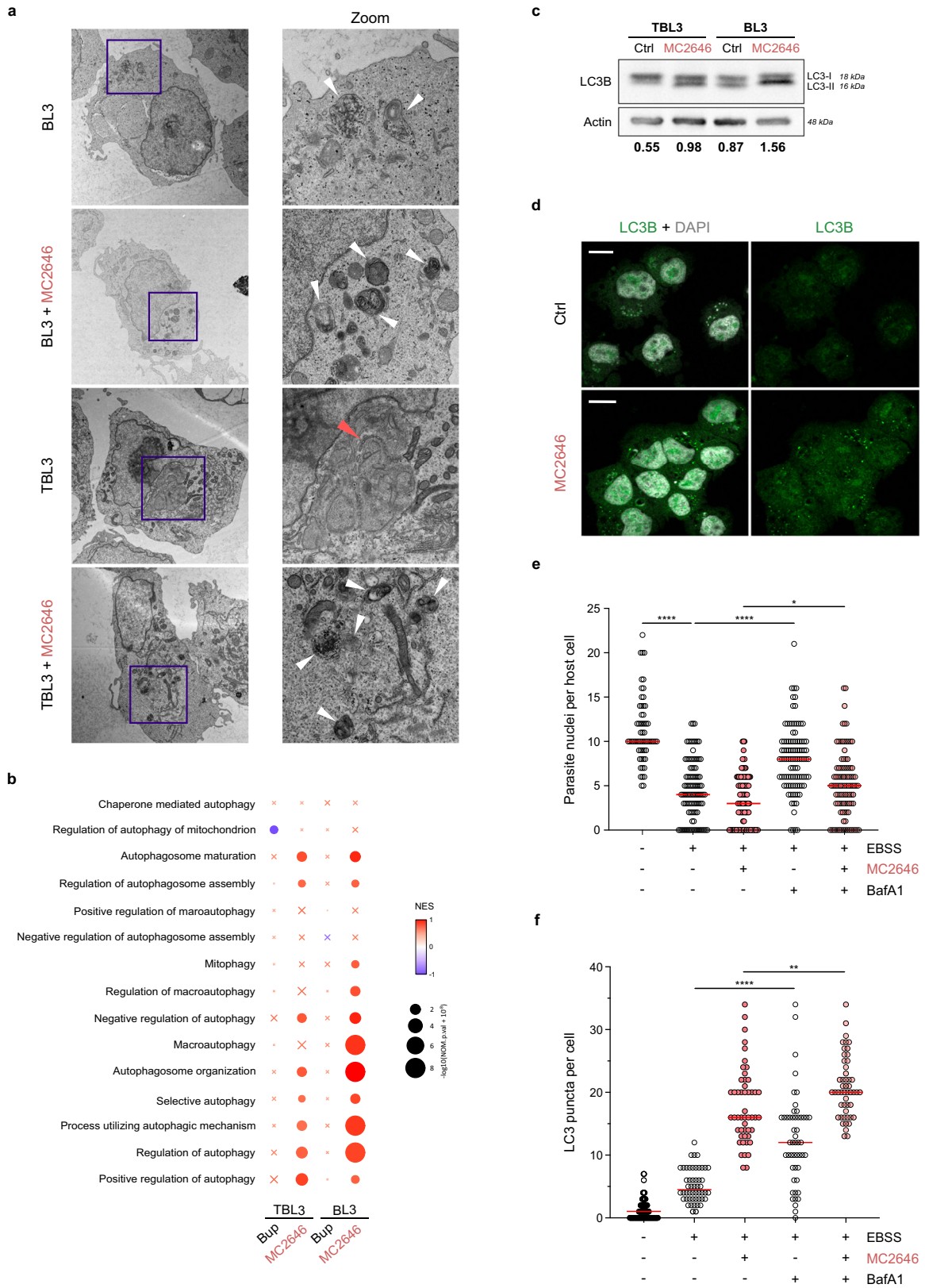

MC2646 compound is an activator of autophagic flux in our cells, whereas Buparvaquone is not. MC2646 treatment led to autophagosome accumulation when fusion with lysosomes was prevented by BafA1. This implies that the induction of autophagy is not due to the decrease in parasite survival, but the contrary. We transfected TBL3 cells with a LC3-RFP-GFP reporter plasmid to follow lysosomal

degradation and autophagosome-lysosome fusion; autophagosomes structures are labelled yellow, whereas fusion with the lysosome creates an acidic pH which quenches the GFP, making autolysosomes appear red. In untreated TBL3 infected cells we observed a perfect colocalization between RFP and GFP [Supplementary Fig. 3e]. Upon MC2646 treatment, the decrease in yellow puncta and the appearance

**Fig. 2 | Treatment with the MC2646 compound restores host cell autophagy.**
**a** Ultrastructural analysis by electron microscopy revealed classical autophagy-related organelle structures in uninfected BL3 cells, but a reduction of these structures in infected TBL3 cells. Treatment with MC2646 restored increased autophagosome in both cells. The electron micrographs are shown at a magnification of 2900x (left) and the highlighted blue box at 9300x (right). The white arrowheads highlight the autophagosome structures in BL3 cells and both cell types treated with MC2646. The red arrow highlights the annulate lamellae visible in infected cells. **b** Gene Signature Enrichment Analysis (GSEA) where color scale represents Normalized Enrichment Score (NES) and circle size represents *p*-value. Gene Ontology Biological Process (GOBP) genesets related to autophagy were used to compare the four RNA-Seq datasets: uninfected BL3 & infected TBL3, treated with Buparvaquone or MC2646, compared to the respective untreated dataset. Crosses indicate enrichment scores that are not statistically significant (FDR *q*-value > 0.25). The GSEA was evaluated with a Weighted Kolmogorov-Smirnov Test. **c** Analysis of LC3B protein in infected TBL3 and uninfected BL3 cells with or without MC2646 treatment. Western blots profiles show LC3-I and LC3-II forms which are restored in TBL3 cells after MC2646 treatment (4 μM, 24 h). The quantification of the ratio between LC3-II:LC3-I is indicated below. Actin was used as a loading control. **d** Infected TBL3 cells were untreated (Ctrl) or incubated with MC2646 (4 μM for 24 h) and LC3B-puncta were monitored by immunofluorescence (green). Host and parasite nuclei are indicated by DAPI staining. Leica microscope x63 magnification; the scale bar corresponds to 15 μm. **e** Autophagy induced by low-nutrient media reduced parasite survival. Infected TBL3 cells were incubated in EBSS media (4 h) and parasite load (determined by number of nuclei per host cell) was monitored. The addition of BafilomycinA1 (BafA1), blocks autophagic flux and rescued parasite survival. At least 50 host cells were counted per condition. Results are significant under Kruskal-Wallis followed by a Dunn's multiple comparison test *p < 0.0191; ****p < 0.0001. **f** Quantification of LC3B puncta (same conditions as Fig. 2e). In all experiments cells were incubated with 50 ng/ml Buparvaquone or 4 μM MC2646 for 24 h. Results are representative of 3 independent experiments. Statistical analysis Dunnett's multiple comparison test. *p < 0.1, ****p < 0.0001.

of only red dots, indicated that autophagosome and lysosome fusion occurred and therefore that autophagic flux is functioning [Supplementary Fig. 3e]. This experiment was technically challenging, due to the difficulty of transfecting parasitized cells. We therefore chose to confirm a more general role of MC2646 as an inducer of autophagic flux using the established experimental system in U2OS cells stably expressing the LC3-RFP-GFP reporter. Treatment with MC2646 resulted in reduced yellow puncta and the appearance of red spots, indicating the compound induces autophagy in other cellular contexts and is not limited to parasite infection [Fig. 3f-g]. These combined experiments support a general role of MC2646 in inducing autophagy that results in parasite elimination.

### eIF5A is sequestered by *Theileria* parasites and released upon MC2646 treatment

To study the mechanisms by which the MC2646 compound induces autophagy and eliminates parasites, we investigated its localization in the cell. Using 'click chemistry' we associated MC2646 with an alkyne (the derivative compound MC4404) bound to fluorescent azide [Supplementary Fig. 1c] and observed that the 'click' drug targets structures adjacent to the parasite nuclei on the macroschizont surface [Fig. 4a]. We were unable to determine exactly the identity of these drug-target structures. When we performed the same experiment after MC2646 treatment, the MC4404 'click' compound marked the remaining parasites, forming a ring around shrunken schizont [Fig. 4a]. The MC4404 compound was toxic in cells and was therefore used as a probe to mark the targeting of the MC2646 drug to the parasite surface. TBL3 cells were incubated with or without the MC2646 autophagy inducer and MC4404 was just added for 30 min for the visualization [Fig. 4a and Supplementary Fig. 4]. A control experiment with a 'click' derivative of a related Tranylcypromine (TCP) compound did not localize to the parasite, demonstrating the specificity of the MC2646-MC4404 localisation [Fig. 4b]. We performed a pull-down experiment using the MC4404 derivative compound followed by mass spectrometry to identify associated proteins. We identified 146 proteins, which we speculated were targets of MC2646 and their associated partners [Supplementary Data 1]. Interestingly, the vast majority of these are bovine proteins, suggesting that the drug is perhaps targeting a sequestered host protein. Pioneering studies have identified schizont-associated host proteins using proximity labelling methodologies[15,16,34]. We compared our dataset with the list of 67 proteins reported to localize near the schizont[15] and proteins enriched in annulate lamellae structures[35,36] which are recruited to the parasite surface. This comparison identified a single protein, the translation factor eIF5A [Supplementary Fig. 4a]. Interestingly, eIF5A is the only protein known to be post-translationally modified by hypusination which is key to its role in regulating the translation of some autophagy proteins, such as TFEB[37], and anti-microbial responses[38]. We

performed immunofluorescence experiments and found that the host protein eIF5A was localized in spots in the vicinity of the schizont structure [Fig. 4c]. However, the eIF5A labelling did not colocalize with the click-drug MC4404 [Fig. 4c], nor with a marker (mab414) of annulate lamellae [Supplementary Fig. 4b]. Co-staining with an antibody recognizing the parasite macroschizont surface (mab1C12) and 3D imaging reconstruction further demonstrated that the eIF5A protein is recruited to the parasite structure [Fig. 4c-d]. Upon treatment with MC2646, eIF5A was released into the cytoplasm of the host cell [Fig. 4c]. This effect was specific to the MC2646 autophagy-inducing drug, as Buparvaquone treatment reduced parasite numbers, but eIF5A remained localized to the residual schizont structure [Supplementary Fig. 5a]. These results suggest that eIF5A is sequestered by intracellular parasites and raised the question whether the release of eIF5A by MC2646 could contribute to autophagy and parasite elimination.

Hypusination of eIF5A is critical for its autophagic function and is mediated by the Deoxyhypusine synthase (DHPS) enzyme. We observed that eIF5A is hypusinated in our cells [Fig. 4e] and that treatment with the DHPS inhibitor GC7, reduced eIF5A hypusination leading to reduced levels of autophagic TFEB protein, a known eIF5A translation target [Fig. 4e and Supplementary Fig. 5d]. MC2646 treatment did not significantly impact hypusinated eIF5A levels [Fig. 4e]. Treatment of infected cells with GC7 had no effects of on parasite survival [Fig. 4f], but increasing GC7 concentrations rescued the parasite survival upon MC2646 treatment [Fig. 4f and Supplementary Fig. 5b]. The reduced parasite survival induced by the AMPK activator 991 or by EBSS, was also rescued by inhibiting hypusination with the GC7 inhibitor [Supplementary Fig. 5c], suggesting that eIF5A hypusination is downstream of the AMPK pathway. Treatment with GC7 blocked autophagy LC3-puncta formation induced by MC2646 [Fig. 4g]. These results suggest that active hypusinated eIF5A is essential for autophagy and parasite clearance induced by MC2646 (or other autophagic inducers). To further demonstrate the importance of eIF5A, we created stable cell lines with depleted eIF5a (sh_eIF5A). Knocking down eIF5A correlated with reduced levels of autophagy-related targets TFEB and ATG3 [Fig. 4h and Supplementary Fig. 5d]. We treated knockdown (sh_eIF5A) and control (sh_Ctrl) cells with MC2646 and monitored parasite load and LC3 puncta. The reduced eIF5A rescued parasite survival upon MC2646 treatment, as measured by parasite nuclei numbers, [Fig. 4i] and blocked the drug-induced LC3 puncta and autophagosome formation [Fig. 4j]. These results demonstrate that host eIF5A is required for the activation of autophagic flux and parasite clearance by the MC2646 drug.

### Discussion
Parasite infections are responsible for a considerable disease burden for livestock, driving the need for better treatments and insight into

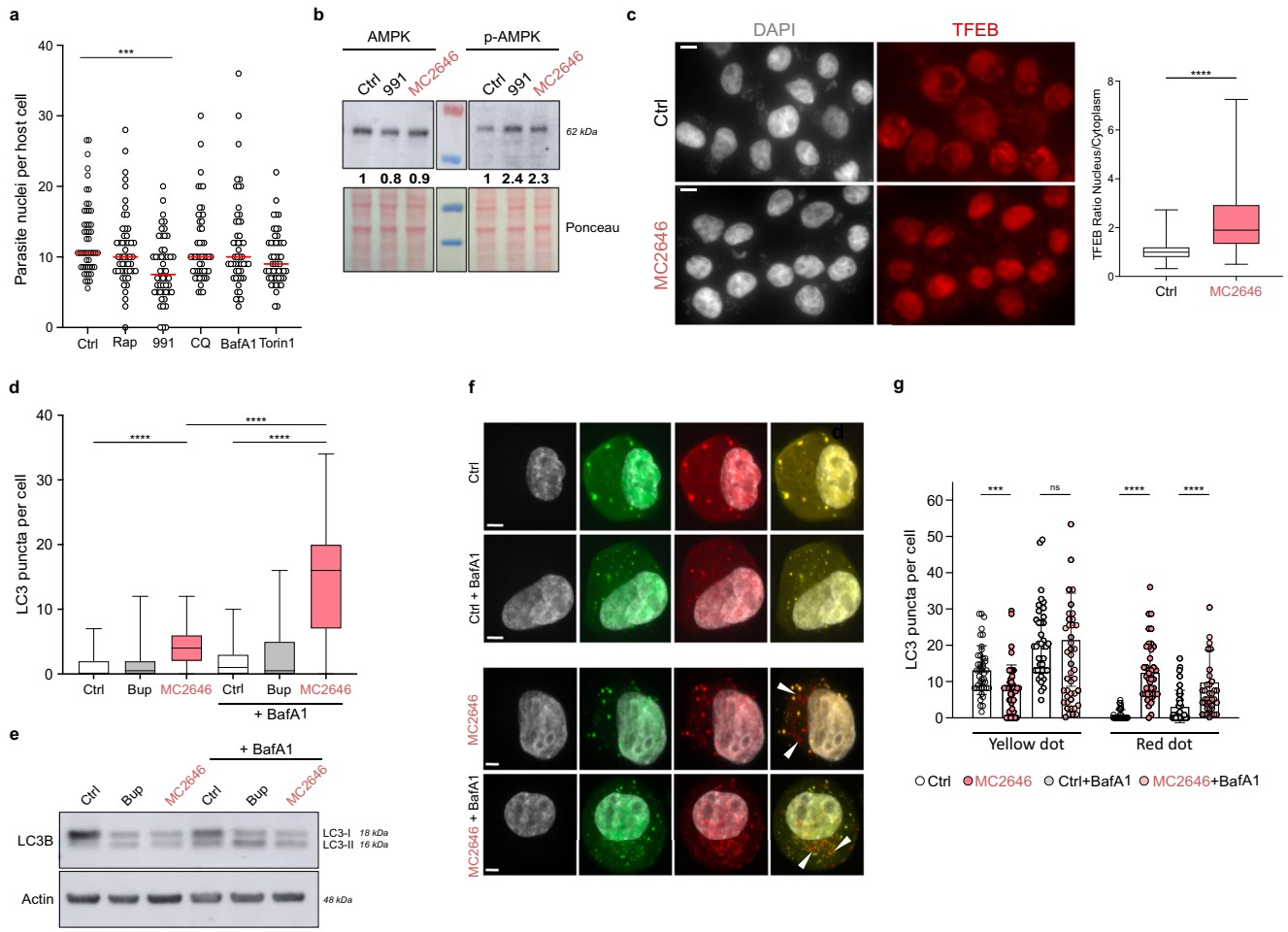

**Fig. 3 | The compound MC2646 induces autophagy flux leading to parasite loss.**
**a** Testing regulators of autophagy: infected TBL3 cells were incubated with Rapamycin (2 μM), 991 (4 μM), CQ (1 μM), BafA1 (1 μM), or Torin1 (1 μM) for 24 h. Parasite survival was monitored by counting nuclei in DAPI-stained images. At least 50 host cells were counted. Statistical 2-way Anova Dunn's multiple comparison test. ***$p = 0.0009$. **b** Treatment of infected TBL3 cells with MC2646 or 991 compounds resulted in activation of the AMPK pathway (phosphorylation of Thr172, p-AMPK) detected by Western blot analysis. The relative quantification levels are indicated. **c** Infected TBL3 cells were treated with MC2646 and nuclear translocation of the transcription factor TFEB was monitored by immunofluorescence. The Nuclear:Cytoplasmic ratio was quantified for $n = 50$ cells. Statistical significance Mann-Whitney test two-tailed ****$p < 0.0001$. Leica microscope x63 magnification; the scale bar corresponds to 15 μm. **d** MC2646 induces LC3B puncta and autophagosomes formation. Infected TBL3 cells were incubated with Buparvaquone or MC2646 (24 h) and LC3B puncta were monitored by immunofluorescence. Bafilomycin A1 (BafA1) was added (50 nM for 3 h) to block the autophagic flux. Statistical significance One way-Anova, Dunnett's multiple comparison test, ****$p < 0.0001$. **e** The MC2646 compound induces autophagic flux. LC3B isoforms were monitored

(same conditions as Fig. 3d) by Western blot analysis. Quantification corresponds to the LC3-II:LC3-I ratio standardized to the Actin control. **f** The MC2646 compound induces autophagic flux and fusion of autophagosomes and lysosomes. Immunofluorescence analysis of U2OS cells stably expressing a LC3-RFP-GFP reporter treated or not with MC2646 for 24 h. Bafilomycin A1 was added (50 nM for 3 h). The red dots, indicating autophagosome induction by the MC2646 compound, are highlighted with white arrows. This is a representative experiment of the 3 that are shown quantitatively in Fig. 3g. Leica microscope x100 magnification; the scale bar corresponds to 5 μm. **g** Human uninfected U2OS cells expressing the LC3-RFP-GFP reporter plasmid treated with MC2646 and/or BafA1. Yellow puncta indicated the absence of autophagic flux, and red puncta fusion between autophagosomes and lysosomes. Statistical significance 2-way Anova Tukey's multiple comparison test. At least 50 host cells were counted per condition. Error bar corresponds to the mean with SD. ***$p < 0.0002$; ****$p < 0.0001$. In all experiments, cells were incubated with 50 ng/ml Buparvaquone or 4 μM MC2646 for 24 h and/or BafA1 (50 nM for 3 h). Results are representative of 3 independent experiments. The boxplots in graphs indicate the 25% (bottom), 50% (center) and 75% quartiles (top). Whiskers represent the minimum (bottom) and the maximum (top).

host-parasite interactions; the cost of *T. parva* alone is >300 M$, with over 1 million cattle deaths per year in sub-Saharan Africa. Host cell autophagy mechanisms function as a defense against intracellular pathogens and must be circumvented for pathogen survival[17,18,39]. Indeed, autophagic processes have been highlighted as part of the dynamic host-parasite interactions in several apicomplexan parasites, such as *Plasmodium* and *Toxoplasma*[20,40–42]. *Theileria* parasites differ from other related parasites by the lack of a parasitophorous vacuole membrane and the direct contact with the host cytoplasm[43]. Here, we show that recruitment of the host eIF5A translation factor may be a novel strategy for hijacking the host autophagic defenses and allowing

parasite survival. This is part of a growing number of sequestered proteins (including IKK, CLASP, JNK2) that contribute to the manipulation of host processes[14,15,44]. Related *Plasmodium* parasites recruit host LC3 to the PVM via the parasite UIS3 protein[19,45]. There is no evidence for a UIS3 homolog in *Theileria* and the C4 drug that targets the UIS3-LC3 interaction was toxic in our cells. Further studies will be necessary to discover the mechanism by which *Theileria* parasites sequester eIF5A and the proximity-labelling techniques used for CLASP1 would be a promising approach, especially as eIF5A was in this sequesterome dataset[15]. eIF5a sequestration could be a strategy employed by other intracellular pathogens, as hypusination is a critical

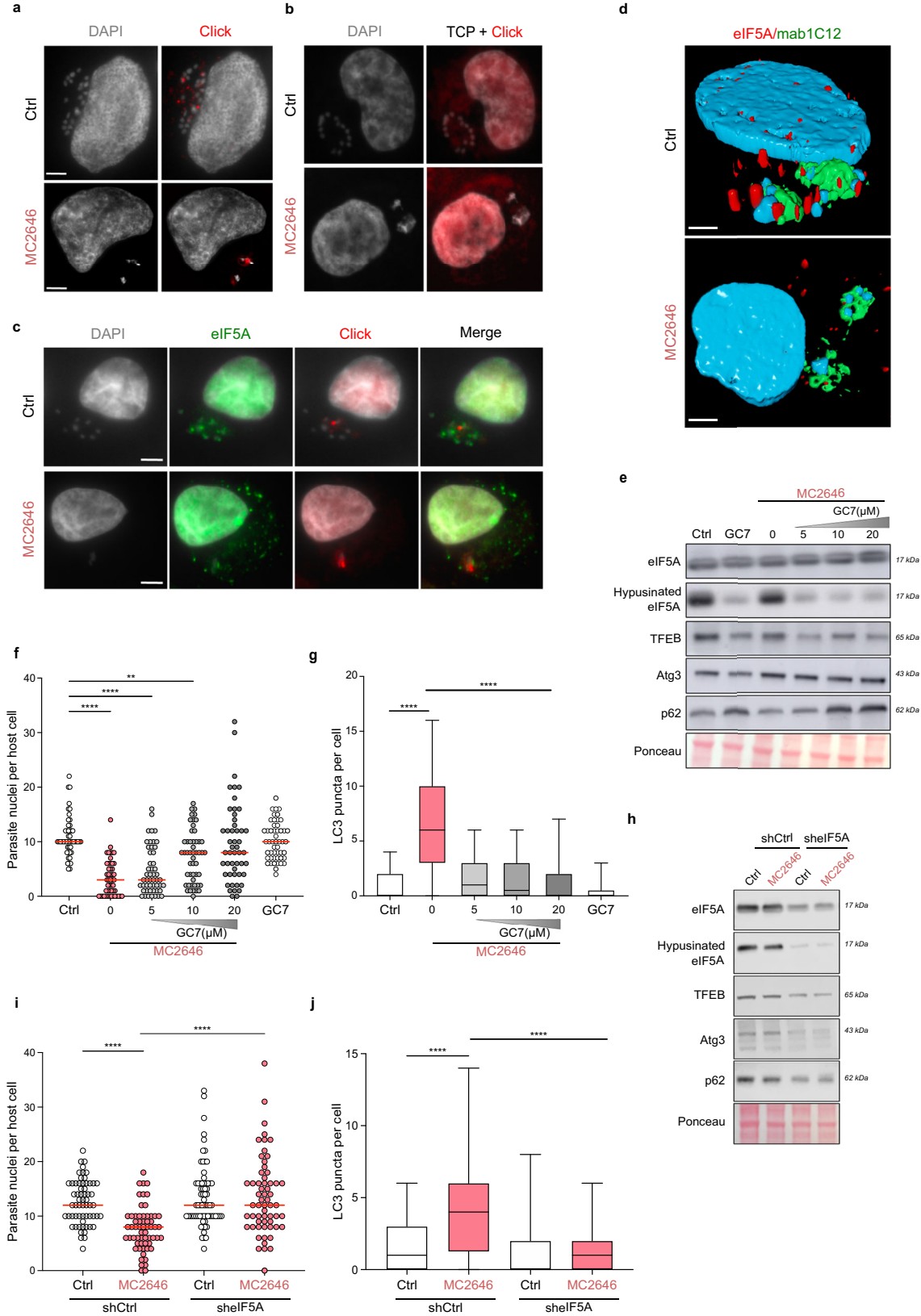

hallmark of the host defense against microbial infection[38]. Further studies should aim to identify the structures that sequester the eIF5A protein. The MC2646 compound provides a powerful tool to explore this further, as the drug releases eIF5A to drive autophagy and parasite clearance. Additional studies will be necessary to define its precise mode of action and breadth of function of the MC2646 compound.

While we observed a release of eIF5A and a rescue by the GC7 compound that inhibits eIF5A hypusination, it remains to be determined exactly how the new drug induces autophagy. Surprisingly, we observed changes in TFEB levels, a target of eIF5A-regulated translation, but not in other autophagic mediators such as Atg3 or p62. The MC2646 compound (and related MC2645 and MC3205 compounds)

**Fig. 4 | The host eIF5A protein is sequestered at the parasite surface and released upon MC2646 treatment. a** The MC2646-click compound (MC4404) is localized in the vicinity of the parasite nuclei in the schizont. Microscopic images of infected TBL3 cells showing immunofluorescence of the MC2646 click compound (MC4404, Azide-Alexa Fluor 594, red) and DAPI-stained nuclei, with or without (Ctrl) MC2646 treatment. Leica microscope x100 magnification; the scale bar corresponds to 5 μm. **b** A control TCP-click compound did not localize to the parasite surface in the presence of absence of the MC2636 treatment. Microscopic images of infected TBL3 cells showing immunofluorescence of the Tranylcypromine click compound (Azide-Alexa Fluor 594, red) and DAPI-stained nuclei, with or without (Ctrl) MC2646 treatment. Leica microscope x100 magnification; the scale bar corresponds to 5 μm. **c** MC2646 treatment released eIF5A from the parasite schizont. We performed immunofluorescence analysis to localize eIF5A (Alexa Fluor 488, green) adjacent to parasite nuclei (DAPI), distinct from MC2646 click MC4404 (Azide-Alexa Fluor 594, red). The eIF5A was released from the schizont upon MC2646 treatment. Leica microscope x100 magnification; the scale bar corresponds to 5 μm. **d** Reconstructed 3D microscopy images of infected TBL3 cells stained with mab1C12 to mark the surface of the parasite macroschizont structure (green) and a specific anti-eIF5A antibody (red) in cells treated or not with MC2646 compound. The parasite and host nuclei are marked by DAPI staining (blue). The host eIF5A protein is recruited to the parasite surface in TBL3 cells and released upon MC2646 drug treatment. Leica microscope x100 magnification; the scale bar corresponds to 5 μm. **e** Inhibition of eIF5A hypusination. Infected TBL3 cells were treated with MC2646 with or without increasing concentration of the DHPS inhibitor GC7 and analysed by Western blot to monitor eIF5A, hypusinated eIF5A, TFEB, ATG3, and p62 proteins. Decreased eIF5A hypusination led to reduced TFEB and Atg3 levels and p62 accumulation. **f** Inhibition of eIF5A hypusination rescued

parasite survival. Infected TBL3 cells were subjected to the above conditions (Fig. 4c) and monitored for parasite load by counting parasite nuclei per cell. At least 50 host cells were counted per condition ($n = 3$). Statistical significance One way-Anova, Dunnett's multiple comparison test, **$p < 0.014$; ****$p < 0.0001$. **g** Inhibition of eIF5A hypusination reversed autophagic flux. Infected TBL3 cells were subjected to the above conditions (Fig. 3c) and monitored for LC3B puncta by immunofluorescence. At least 50 host cells were counted per condition. Results are significant under Kruskal-Wallis followed by a Dunn's multiple comparison test ****$p < 0.0001$. **h** Effect of eIF5A depletion. Tac12 infected macrophages expressing stable sh_eIF5A (compared to control sh_Ctrl) were analysed by Western blot upon MC2646 treatment. eIF5A knockdown resulted in reduced levels of TFEB, ATG3, and p62. At least 50 host cells were counted per condition. Results are significant under Kruskal-Wallis followed by a Dunn's multiple comparison test ****$p < 0.0001$. **i** eIF5A depletion rescued parasite survival. Tac12 cells expressing sh_eIF5A showed restored parasite load (parasite nuclei per host cell) upon MC2646 treatment. At least 50 host cells were counted per condition. Results are significant under Kruskal-Wallis followed by a Dunn's multiple comparison test ****$p < 0.0001$. **j** eIF5A depletion blocked autophagosome formation. Immunofluorescence analysis of Tac12 cells expressing sh_eIF5A showed reduced LC3B puncta in the presence of MC2646 drug. At least 50 host cells were counted per condition. Results are significant under Kruskal-Wallis followed by a Dunn's multiple comparison test ****$p < 0.0001$. In all experiments, cells were incubated with 50 ng/ml Buparvaquone or 4 μM MC2646 for 24 h and/or BafA1 (50 nM for 3 h). Results are representative of 3 independent experiments. Statistical analysis Dunnett's multiple comparison test. **$p < 0.01$, ****$p < 0.0001$. The boxplots in graphs indicate the 25% (bottom), 50% (center) and 75% quartiles (top). Whiskers represent the minimum (bottom) and the maximum (top).

are derivatives of Tranylcypromine which is currently used clinically to treat depression, suggesting that they will have favorable pharmaco-profiling as drugs. Interestingly, MC2646 is also the first drug to uncouple *Theileria* parasite survival from host cell survival, challenging the notion of 'parasite addiction' suggested by studies with Buparvaquone[12]. MC2646 represents a promising drug to treat Buparvaquone-resistant strains as the mechanisms of action appear distinct. It could also provide a tool to explore pathogen subversion of autophagy in other models and could serve as an autophagy-inducing drug to treat infection and diseases.

# Methods

## Cell culture

All bovine cell lines were previously documented. TBL3 cells were derived from in vitro infection of the spontaneous bovine B lymphosarcoma cell line, BL3, with Hissar stock of *T. annulata*. Tac12 is a line of *T. annulata*-infected bovine macrophages. The TpMD409 lymphocyte cell line is infected with *T. parva*. Cells were cultured in RPMI 1640 (Gibco-BRL), supplemented with 10% heat-inactivated fetal bovine serum (FBS), 4 mM L-glutamine, 25 mM HEPES, 10 mM β-mercaptoethanol and 100 mg/ml of penicillin/streptomycin in a humidified 5% CO$_2$ atmosphere at 37 °C. G. Langsley (Institut Cochin, Paris, France) provided the TBL3, BL3 and TpMD409 cell lines and K. Woods (University of Bern, Switzerland) provided the Tac12 cell line.

## Drug treatment of infected cells

Bovine cell lines were treated with drugs under the following conditions: Buparvaquone was used at 50 ng/ml for 24 or 48 h; compound MC2646 was used at 1 μM for 48 h or 4 μM for 24 h; the AMPK activator (991) was used at 4 μM for 24 h; the AMPK inhibitor (SBI) was used at 1 μM for 24 h; the hypusination inhibitor, GC7, was used at 10 μM for 24 h; chloroquine (CQ) and Torin1 were used at 1 μM for 24 h; Bafilomycin A1 (BafA1) 1 μM for 24 h or 50 nM for 3 h.

## Merogony induction

Macroschizont-infected Tac12 cells were induced to differentiate to merogony by increasing the culture temperature to 41 °C. Cells were passaged each time they reached confluence and $2 \times 10^6$ cells collected

at day 0 (macroschizont stage) and day 8 (merogony stage) for RNA extraction and $4 \times 10^3$ cells for immunofluorescence at the same time-points.

## Microscopic screening

Drug screening was carried out as previously described (*Communications Biology*, in press) Tac12 infected macrophages expressing a GFP-CLASP fusion protein[15] were plated in 96-well plates and treated with the compounds at 10 μM for 48 h. Immunofluorescence was performed on fixed cells using a specific anti-H3K18me1 antibody to label parasite nuclei. Cells were incubated with DAPI to detect host and parasite nuclear DNA. The parasite surface membrane was monitored by GFP-CLASP fluorescence[15]. Image capture (30 fields per condition) and analysis was performed with the Opera Phenix microscope (Perkin Elmer, Photonic BioImaging platform, Pasteur Institute) and the associated Acapella Software to monitor host and parasite survival.

## Western blot analysis

$5 \times 10^5$ cells were cultured in 12-well plates for 24 h before the extraction. Proteins were extracted with Laemli lysis buffer, then resolved by running gels SDS-PAGE 4-12%, and transferred to nitrocellulose membranes, before being incubated overnight at 4 °C with primary antibodies (mouse or rabbit) against eIF5A (BD Biosciences, ref611976, 1/10000), eIF5A-hypusinated (EMD Millipore Corp, ABS1064, 1/4000), AMPK (Cell Signaling, 2532 S, 1/2000), 755 pAMPK (Cell Signaling, 2531 S, 1/2000), LC3B (Abcam, ab51520, 1/2000), p62 (Abcam, ab56416, 1/2500), ATG3 (Abcam, ab108251, 1/2000), TFEB (Proteintech, ref 13372, 1/2000) and Actin (Sigma, A1978,1/10000), followed by secondary antibodies produced against mouse or rabbit antibodies. The phosphorylated AMPK antibody was diluted in TBST and the cells were treated with a cocktail of protease inhibitors. [See Supplementary Table 2 for details of antibodies and dilutions].

## Immunofluorescence and microscopy

B cells (BL3 & TBL3) were washed with PBS and $3 \times 10^4$ cells per slide were centrifuged with Cytospin (10 min at 1000 × g) to stick to the blade. Cells were fixed in 3.7% paraformaldehyde for 10 min, or with cold 100% Methanol for 5 mins; then permeabilized in 0.2% Triton-PBS

for 10 min. Cells were blocked with PBS 0.2% Tween (PBST)−1% BSA for 30 min. The primary antibodies were diluted in PBST and incubated for 1 h at the following dilutions: The primary antibodies were diluted in PBST and incubated for 1 h at the following dilutions: eIF5A (Abcam, ab137561, 1/200); TFEB (1/300); p62 (Abcam, ab56416, 1/500); LC3B (Proteintech, ref 18725, 1/250) and mab414 (Abcam, 24609, 1/500). Cells were washed three times 766 with PBST and incubated with secondary antibodies for 30 min at 1:1000 dilution (Invitrogen). Cells were washed three times with PBST and finally mounted on glass coverslips, adding mounting medium containing DAPI. Samples were analyzed using a Leica DMI 6000 epifluorescence microscope. Images were generated and processed using Metamorph and ImageJ software. Parasite counting experiments were performed with a minimum $n = 30$ cells. [See Supplementary Table 2 for details of antibodies and dilutions]. For electron microscopy, cells were fixed with 1% glutaraldehyde/2% paraformaldehyde for at least 1 h. After PBS washes, Tac12 cells were scrapped and pelleted in 1% agarose. BL3 suspension cells were also pelleted in 1% agarose after washing in PBS. Small blocks of agarose-embedded cells were post-fixed with 1% osmium tetroxide with 1.5% potassium ferrocyanide in PBS (pH 7.4), progressively dehydrated in ethanol, infiltrated and embedded (60 °C for 24 h) in low-viscosity epoxy resin (Agar Scientific Ltd). 70-nm-thick sections were cut using an EM UC6 ultramicrotome (Leica), mounted on copper grids, and stained with uranyl acetate and lead citrate. Sections were examined with a 120 kV TEM (Tecnai 12, Thermo Fischer Scientific) and imaging was done with a 4 K CDD camera (Oneview, Gatan).

### Neon-transfection

Cells were plated at $5 \times 10^5$ cells per well (24-well plate), rinsed with PBS, then centrifuge for 5 min at 900 rpm, and add 10 µL of Buffer R on the cell pellet, from the Neon-transfection kit. Then 1.5 µL of siRNA at 20 µM (i.e. 400 nM) or 0.5 µg DNA (plasmid) were added to the cell pellet. Buffer E was put in an electrolyte tube and inserted into the machine. Impulse conditions voltage for infected lymphocytes were 1200 V/40 ms/1 pulse.

### Click chemistry

To follow MC2646 localisation and targeting, we used a derived compound adapted to the "Click" reaction. The MC2646 coupled with an alkyne, gives the compound MC4404. The azide can be coupled with an AlexaFluor-488 or 594 nm, or with a Biotin. We treated cells for 30–60 min with 1 µM of MC4404. After PFA fixation of cells on slides, permeabilization and blocking (see immunofluorescence protocol), we realize the "click reaction". The reaction buffers of the "Click-IT Plus EdU Alexa fluor 594 Imaging kit", from Invitrogen, consisted of two reaction buffers, copper and AlexaFluor594-Azide. Copper makes it possible to bind the compound with the fluorescent or the biotin. We visualized the localization of the molecule and its target in the cell, after washing and mounting with a medium containing DAPI. A second "clickable and photo-crosslinkable" compound (called MC4564) allowing target recognition and solid interaction, after UV exposure (300 nm, 2 min exposure), between the target and the compound. We treated the cells with 10 µM MC4564, 30–60 min, then we performed a UV "crosslink", extracted the proteins with RIPA buffer, then incubated the proteins with the click mix where the alkyne is combined with a Biotin and a fluorescent (TAMRA biotin-Azide, from Click chemistry tools). The protein extract and the biotinylated compound, following the click reaction, were incubated overnight with magnetic beads combined with streptavidin, allowing the biotin-streptavidin interaction. After 3 washes with RIPA buffer and 3 washes with water, some of the beads are taken up in Laemli buffer to visualize target proteins by fluorescence on gel. The other half was taken up in water for a Mass Spectrometry (MS) analysis. We defined 3 conditions for the MS experiment on macrophages infected with T. annulata, Tac12: cells were treated with the 'unclickable' compound MC2646, 1 h at 10 µM, as

negative control; MC4564, 1 h at 10 µM, to identify potential targets; cells treated for 1 h with 10 µM of MC4562 and 50 µM of MC2646, to create a competition condition and increase the selectivity of potential candidates.

### RNA extraction and RT-qPCR

Total RNA was extracted using a Nucleospin RNA extraction kit (MachereyNagel) following the manufacturer's protocol. 1 µg of total RNA was reverse transcribed with Superscript III Reverse transciptase Kit (Invitrogen). Real-time quantitative PCR was performed to analyse relative gene expression levels using SyberGreen Master Mix (Applied Biosystem) following the manufacturer's protocol. Relative expression values were normalized with housekeeping gene mRNA HSP70 or Actin. Primer sequences are listed in Supplementary Table 1.

### Bioinformatic RNA-Seq analysis

$5 \times 10^6$ cells (BL3 or TBL3) were used as starting material to extract RNA using the TRI-reagent (SIGMA, T-9424) protocol. The RNA-Seq data were analysed on the cluster of French Institute of Bioinformatics (IFB), using the workflow developed by the BiBs platform (version 0.5) and based on RASflow[46], which integrates all the following steps. Trimming of adapters and low-quality reads was done by Trim Galore! and we used the HISAT2 aligner[47] to map the reads on the bovine genome (assembly accession GCF_002263795.1 for Bos Taurus). The resulting mapped reads were assigned to genomic features ("gene" parameter) using featureCounts on a similarly fused GTF annotation file. Finally, the differential expression analysis (DEA) was carried out using DESeq2[48]. The two datasets were analyzed separately. The exact configurations for the workflow can be found attached. Gene set enrichment analysis (GSEA)[49] was carried out using the version v4.2.3. windows application. The gene counts produced by RASflow were used as input for the GSEA of each dataset. The geneset database was created by subsetting MSigDB v7.4, for autophagy-related genesets. The Bovine.chip file, as well as the parameters for each comparison (.rpt) can be found attached.

### Construction of shRNA Tac12 cell lines

shRNA oligos were designed based on initial siRNA test results. The designed oligos were resuspended in ddH2O and incubated in 0.5x annealing buffer. In parallel, the vector was digested using AgeI and EcoRI high-fidelity enzymes, this was followed by gel purification. Annealed shRNA oligos were ligated with the gel-purified open vector using T4 DNA ligase. These ligation reaction products were then used to transform competent StBl3 cells. Individual Amp-resistant colonies were selected and DNA was extracted using the Qiagen Kit. A XhoI digestion was set up on the extracted DNA samples to screen for the samples that have the shRNA insert in the vector. Successful clones were sequenced to verify the identity of the shRNA insert and used to make lentiviruses. A similar approach was used to clone the shRNA oligos into the pLKO GFP vector. Construction on a stable cell line Tac12 stable sh_eIF5A that stably expressed the sh-eIF5A and a GFP tag. Lentiviruses were produced in HEK-293T cells and Tac12 cells were infected with HEK-293T supernatant containing lentiviruses particles. Infection efficiency was assayed according the percentage of GFP positive Tac12 cells. Knockdown cells were compared to empty vector sh_Ctrl control cell lines. Oligonucleotide sequences are listed in Supplementary Table 1.

### Mass spectrometry analysis for drug target identification

A second "clickable and photo-crosslinkable" compound (called MC4564) was designed to enable target recognition and solid interaction, after UV exposure, between the target and the compound. Cells were treated with 10 µM of MC4564 for 30–60 min, followed by UV cross-linking (300 nm, 2 min exposure). Cells were then lysed with RIPA buffer, and the whole protein extract was incubated with click

mixture where the alkyne group is coupled with a biotin and a fluorophore (TAMRA 800 biotin-Azide, from Click chemistry tools). The solution was incubated overnight with streptavidin-coupled magnetic beads (Streptavidin Dynabeads™ MyOne™ C1, Invitrogen), allowing biotin-streptavidin interaction. After 3 washes with RIPA buffer and 3 washes with water, half the beads were removed to visualize compound-protein interaction by fluorescence on a gel. The other half were resuspended in water for analysis by mass spectrometry (MS). The 3 conditions for the MS experiment were (i) Tac12 cells treated with the "non-clickable" compound MC2646 (10 μM, 1 h) as a negative control, (ii) Tac12 cells treated with MC4564 (10 μM, 1 h), to identify potential targets, and (iii) Tac12 cells treated for 1 h with 10 μM of MC4562 and 50 μM of MC2646, to create competition between nonclickable and clickable compounds, and thus increase the specificity of potential candidates. Beads were incubated overnight at 37 °C with 20 μl of 25 mM $NH_4HCO_3$ buffer containing 0.2 μg of sequencing-grade trypsin. The resulting peptides were loaded and desalted on evotips provided by Evosep (Odense, Denmark) according to manufacturer's procedure.

### LC-MS/MS acquisition
Samples were analyzed on an Orbitrap Fusion mass spectrometer (ThermoFisher Scientific, Waltham, MA, USA) coupled with an Evosep one system (Evosep, Odense, Denmark) operating with the 30SPD method developed by the manufacturer. Briefly, the method is based on a 44-min gradient and a total cycle time of 48 min with a C18 analytical column (0.15 × 150 mm, 1.9 μm beads, ref EV-1106) equilibrated at room temperature and operated at a flow rate of 500 nl/min. H20/0.1 % FA was used as solvent A and ACN/ 0.1 % FA as solvent B. The mass spectrometer was operated by data-dependent MS/MS mode. Peptide masses were analyzed in the Orbitrap cell in full ion scan mode, at a resolution of 120,000, a mass range of m/z 350-1550 and an AGC target of 4.105. MS/MS were performed in the top speed 3 s mode. Peptides were selected for fragmentation by Higher-energy C-trap Dissociation (HCD) with a Normalized Collisional Energy of 27% and a dynamic exclusion of 60 s. Fragment masses were measured in an Ion trap in the rapid mode, with and an AGC target of 1.104. Monocharged peptides and unassigned charge states were excluded from the MS/MS acquisition. The maximum ion accumulation times were set to 100 ms for MS and 35 ms for MS/MS acquisitions respectively.

### MS data analysis
Label Free quantitation was performed using Progenesis QI for proteomics software version 4.2 (Waters, Milford, MA, USA). The software was allowed to automatically align data to a common reference chromatogram to minimize missing values. Then, the default peak-picking settings were used to detect features in the raw MS files and a most suitable reference was chosen by the software for normalization of data following the normalization to all proteins method. A between-subject experiment design was chosen to create groups of biological replicates. MS/MS spectra were exported and searched for protein identification using PEAKS STUDIO Xpro software (Bioinformatics Solutions Inc., Waterloo, ON, Canada). De Novo was run with the following parameters: trypsin as enzyme (specific), half of disulfide bridge (C) as fixed and deamidation (NQ)/oxidation (M)/phosphorylation (STY) as variable modifications. Precursor and fragment mass tolerances were set to respectively 15 ppm and 0.5 Da. Database research was conducted against a combined database of UniprotKB Bos Taurus (release 2021, 37512 entries) and Theileria Annulata (release 2021, 7510 entries). The maximum of variable PTM and missed cleavages per peptide were set to 2. Spectra were filtered using a 1% FDR at peptide level. Identification results were then imported into Progenesis to convert peptide-based data to protein expression data using the Hi-3 based protein quantification method. Log2 transformed data were finally used for statistical analysis i.e.

evaluation of differentially present proteins between two groups using a Student's bilateral t-test. A p-value better than 0.05 was used to filter differential candidates.

### Reporting summary
Further information on research design is available in the Nature Portfolio Reporting Summary linked to this article.

## Data availability
All data generated or analyzed during this study are included in this published article. The RNA-Sequencing data have been deposited to the GEO database and the accession number is GSE250088. The Mass Spectrometry proteomics data were deposited in the ProteomeXchange Consortium via the PRIDE partner repository with the dataset identifier PXD047937. Source data are provided with this paper.

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

## Acknowledgements

We thank members of the Weitzman laboratory for critical reading of the manuscript and invaluable advice on this study and members of the UMR7216 for helpful discussions. We thank G. Langsley (Institut Cochin, Paris, France) for the generous gift of TBL3, BL3 and TpMD409 cells and K. Woods (University of Bern, Switzerland) for Tac12 cells and the mab414 reagent. We thank Guido Kroemer (Centre de Recherche des Cordeliers, Paris, France), Benoit Violet (Institut Cochin, Paris, France), Etienne Morel (Institut Necker Enfants Malades, Paris, France) and Lucile Espert (IRIM Montpellier, France) for providing reagents and Franck Letourneur for sequencing (GENOM'IC Platform, Institut Cochin, Paris, France). JBW thanks Matthew Weitzman and Claire Gawer for constant advise and support. This work was supported by the French National Research Agency (ANR PATHO-METHYLOME #ANR-15-CE12-0020) and (ANR 20-PAMR-0011 TheraEPI), the EUR G.E.N.E. (#ANR-17-EURE-0013), and the "Who Am I?" Laboratory of Excellence #ANR-11-LABX-0071 funded by the French Government through its "Investments for the Future" program operated by the ANR under grant #ANR-11-IDEX-0005-01, the Fondation ARC pour la Recherche sur le Cancer (ARC n°228308_ARCP, AAP ARC 2020 PJA3), the PARA-SET project funded by IDEX UP AAP EMERGENCE (#IDEX-2021-I-053). The Weitzman laboratory is supported by the Fondation pour la Recherche Médicale (Equipe FRM #EQU202203014701). JBW was a senior member of the Institut Universitaire de France (IUF). JBW and AP are supported by a grant from DIM1HEALTH Région Ile de France. We are grateful to the support of the technical platforms at the Université Paris Cité including EPI2, Epifluorescence Microscopy for Epigenetics, GENOM'IC Platform (Institut Cochin), the Bioinformatics BiBS platform (especially Magali Hennion for advice) and the Imago-Seine core facility (Institut Jacques Monod, member of France-BioImaging #ANR-10-INBS-04 and IBiSA, with the support of Labex "Who Am I", Inserm Plan Cancer and Fondation Bettencourt Schueller).

## Author contributions

JBW developed the concept, provided overall supervision and wrote the manuscript. MV designed the study, performed experiments, analysed the results and wrote the manuscript. MV, JB and AP prepared the figures. NL, IK, JB, AR, AA, YK, and SM performed experiments. AP performed bioinformatics analysis. SV, RF and AM generated the chemical library and modified derivatives. LL, GC and CD performed EM and proteomic analysis.

## Competing interests

All authors do not have any competing interests.
