## [Peer Review File · Nature Communications]

Theileria parasites sequester host eIF5A to escape elimination by host-mediated autophagyREVIEWER COMMENTS

Reviewer #1 (Remarks to the Author):

The interaction between intracellular *Theileria* schizonts and their bovine host cells is unique. *Theileria* is the only eukaryotic cell known to transform their host cells. Little is understood about the molecular mechanisms by which *Theileria* drives transformation, and even less is known about how the parasite escapes attack from the host defence system, in particular autophagy. The data presented in this manuscript contribute greatly to our understanding of this process and I recommend publication after my questions and concerns have been addressed.

In the first part of this study, the authors demonstrate that the compound MC2646 reduces parasite load per cell, without impacting the survival or cell cycle progression of the host cell. Although cell survival was not impacted, the transformed phenotype, as measured by colony formation on soft agar, was inhibited. They next showed that autophagy (LC3 puncta, and LC3-II) is low in infected cells and is induced by treatment with MC2646. They show that TFEB, a transcription factor involved in the regulation of autophagy, is translocated to the nucleus following MC2646 treatment. Using Click chemistry, the authors identify eIF5a as a potential target of MC2646, although an interaction between MC2646 and eIF5a is not verified. Finally, using shRNA to knock down eIF5a expression, and the compound GC7 to inhibit post-translational modification (hypusination) of eIF5a, the authors show that active hypusinated eIF5a is required for autophagy-mediated parasite clearance induced by MC2646.

The methodology is sound and the use of statistics is appropriate. The data presented here are rather complex and I have a few suggestions and questions which I hope will allow the authors to make this work more accessible. In particular, a co-staining of eIF5a and the schizont surface is missing.

Fig 1b. Referring to "parasite nuclei" per host cell rather than "parasites" per cell would be more accurate. The schizont is thought to be a multinuclear syncytium, rather than distinct parasites, so I find "parasite number" misleading.

Fig 1f – the colony forming assay shown here is not very clear. Perhaps this is because I only have a low resolution version of the figures.

Fig 2a. Due to the low resolution of reviewer files I cannot assess how convincing the TEM data is. I would like to see the full resolution figures before commenting on the autophagosome structures induced upon treatment with MC2646. More clear labelling of the parasite would be helpful. And where are the TEM for BL3+ MC2646?

Figure 2a,b,c,d. You show that MC2646 affects host autophagy in BL3 cells (increase in autophagy related gene pathways and an increase in lipidated LC3). Please also show LC3B puncta formation in BL3 cells before and after treatment (and see previous comment about TEM).

How do you explain that rapamycin and torin1 don't reduce parasite number, if they indeed induce autophagy in infected cells? Did you verify that autophagy is induced upon treatment with these classic autophagy activators? Please expand on this.

Figure 2c: western blotting with LC3-I and LC3-II. Can you quantify the relative amounts of LC3-II as you do in figure 3f. This might make the data clearer. I do find the accumulation of LC3-II in MC2646 treated cells compared to the control believable (fig 2c, both BL3 and TBL3), although quantification would help. However I'm a bit lost with figure 3e. Why do you state that MC2646 induces autophagy but buparvaquone doesn't? The ratio of LC3-II:I are the same in both treatments. Please explain.

You show nicely that TFEB nuclear translocation is induced by MC2646 treatment. Did you consider knock out, knock down or inhibition of TFEB? Wouldn't you expect this to block the action of MC2646?

Figure 3f. Please provide a quantification of red/yellow puncta in TBL3 cells following treatment with MC2646 +/- Baf1 to support the data shown in fig 3f. The data shown in the IFA images, particularly the bottom panel with Baf1, don't seem to support the description in the text – I don't see many red puncta in the bottom panel, and in the 3rd panel they are at first glance overwhelmed with the large yellow puncta. Maybe it's worth highlighting the red dots with arrows, or a line in the text. Further, it would be interesting to point out the parasite in these cells. Is LC3 excluded from the nucleus following MC2646 treatment? What does this signify? I find that the grey DAPI signal detracts from the red/green data.

Major point: Figure 4a/b: antibody staining of the schizont surface (TaSP or 1C12) is necessary to show the localization of the drug target as well as eIF5a in infected cells. Why not show a nice 3D reconstruction with 1C12, as you do with mAB414 in figure S3? In figure 4b, the eIF5a staining looks like it overlaps with parasite nuclei rather than the surface of the parasite. How can you exclude that your antibody is cross reacting with Theileria eIF5a?

Using Click chemistry, you show that eIF5a might be a potential target of MC2646. How do you explain the lack of co-localisation? It seems more likely to me that eIF5a is an indirect target of MC2646 – please expand on this.

Line 191-192. You don't show that reduced eIF5a hypusination causes a decrease in TEFB and ATG3, you just show that these two observations correlate with each other. Please modify text. You don't discuss Atg3 expression following MC2646 treatment, with or without GC7. If anything, ATG3 seems to increase following MC2646 treatment, regardless of treatment with GC7. The expression of p62 is not discussed at all and does not seem to be impacted by MC2646 treatment. Have I missed something?

Reviewer #2 (Remarks to the Author):

In this manuscript, Villares and colleagues discovered a novel compound named MC2646 that can effectively suppress the growth of Theileria parasites in host cells. Mechanistically, Theileria parasites suppressed host cell autophagy to facilitate their survival, and MC2646 efficiently induced autophagy probably via activating AMPK to suppress parasites growth. Indeed, MC2646 can be a very useful tool for the autophagy field. The authors also discovered that eIF5A may be sequestered by intracellular parasites, although the physiological impact of such sequestration remains unclear. Pharmaceutical inhibition of eIF5A inhibited autophagy and abrogated MC2646's parasites-inhibition effect.

The following points may help further improve the manuscript:

1. One or two paragraphs of introduction would be very helpful for readers to learn the background knowledge.
2. Line 51 and Fig1a. Although it's published, it is still helpful to show some examples of the screen images of positive and negative controls, to directly show readers how each parameter is measured.
3. Line 65 and Fig 1c: Apoptosis = "sub G1 population"? How this assay of cell cycle and apoptosis was performed? Original images could be shown as an example (in supplementary figures), and more details can be provided in the figure legend.
4. Fig 2d. Do parasites localize within autophagosomes when treated with MC2646? Could these parasites be digested/killed by lysosomes?
5. Fig 2e. MC2646 still reduced parasites load in the presence of BafA1 (comparing the last two columns). Does that suggest that MC2646 may at least partially work in an autophagy/lysosome-independent manner to suppress parasites survival?
6. Fig 3a. Do all of these tested drugs work efficiently to induce or inhibit autophagy as expected? It is confusing why 991 but not Rap or Torin1 can significantly suppress parasites, as rap and torin1 should be potent autophagy inducers as well.
7. Fig 3f. Quantification is required. Also, BafA1 treatment did not increase the number of green/yellow dots (autophagosomes), nor reducing the number of red dots (functional lysosomes)

in control cells. Does BafA1 work well in TBL3 cells?

8. Fig 3g. Again, red dots are active lysosomes. How to explain that BafA1 treatment did not cause reduction of red dots? The autophagy induction by MC2646 is pretty convincing though.

9. Fig 4a. A negative control would be helpful here by adding another alkyne compound, or MC2646, to cells for Click to confirm that the staining is specific. Also, it seems that with the alkyne modification of MC2646 abrogated its inhibitory function because it does not suppress parasite growth in "Ctrl" cells? If that's the case, the interactome of MC2646 could be changed by the modification as well.

10. Fig 4b. eIF5A localizes to ribosomes to facilitate translation, thus shouldn't form large puncta as its functioning form. Could the authors confirm the staining specificity of eIF5A in this figure? It might be hard to directly image ribosomes, but would be helpful if the authors could confirm the finding using biochemical approaches to show that eIF5A is sequestered by parasites, and released to the cytosol upon MC2646 treatment.

11. Fig 4. Generally, this figure shows two very interesting observations: 1) the potential interaction between eIF5A and parasites, and 2) GC7 treatment almost fully blocked MC2646's parasite-suppressing effect. Although MC2646 may not directly interact with eIF5A, several interesting questions still arise: How does the infection alter host translation, including the general translation rate (OPP-Click assay), or the translato~~m~~me in depth (ribosome pull-down sequencing/polysome profiling/ribosomal profiling)? eIF5A may specifically promote translation of hard-to-read regions such as polyprolines. Does the parasite infection lead to translation stalls at those regions?

12. What's the relationship between AMPK and eIF5A? Would GC7 treatment also abolish 991's effects on activating AMPK and suppressing parasites growth?

Minor points:

13. Line 32: It is "unknown"...

14. Line 51: histone "marker"

15. Fig 1a. The control lines are not red as indicated by the legend.

16. It seems that eIF5A is not the major finding in this paper because eIF5A only came up in Fig 4 without a clear mechanism being revealed, thus the short title "Theileria sequesters eIF5a" is not very accurate. Similarly, the main title shall also be adjusted and focus more on the novel autophagy-inducing compound.

NCOMMS-22-35623-T Villares et al. REBUTTAL LETTER TO REVIEWER'S COMMENTS**Reviewer #1**

The interaction between intracellular Theileria schizonts and their bovine host cells is unique. Theileria is the only eukaryotic cell known to transform their host cells. Little is understood about the molecular mechanisms by which Theileria drives transformation, and even less is known about how the parasite escapes attack from the host defence system, in particular autophagy. The data presented in this manuscript contribute greatly to our understanding of this process and I recommend publication after my questions and concerns have been addressed.

We thank the reviewer for these encouraging marks and for carefully evaluating our work.

In the first part of this study, the authors demonstrate that the compound MC2646 reduces parasite load per cell, without impacting the survival or cell cycle progression of the host cell. Although cell survival was not impacted, the transformed phenotype, as measured by colony formation on soft agar, was inhibited. They next showed that autophagy (LC3 puncta, and LC3-II) is low in infected cells and is induced by treatment with MC2646. They show that TFEB, a transcription factor involved in the regulation of autophagy, is translocated to the nucleus following MC2646 treatment. Using Click chemistry, the authors identify eIF5a as a potential target of MC2646, although an interaction between MC2646 and eIF5a is not verified. Finally, using shRNA to knock down eIF5a expression, and the compound GC7 to inhibit post-translational modification (hypusination) of eIF5a, the authors show that active hypusinated eIF5a is required for autophagy-mediated parasite clearance induced by MC2646.

The methodology is sound and the use of statistics is appropriate. The data presented here are rather complex and I have a few suggestions and questions which I hope will allow the authors to make this work more accessible. In particular, a co-staining of eIF5a and the schizont surface is missing.

We endeavored to make the data more accessible in the revised manuscript. We thank the reviewer for pointing out the changes to be made. We have now included a figure of the co-staining of eIF5a and the parasite schizont surface. The new Figure 4d shows a 3D reconstruction of confocal microscopy analysis using an antibody recognizing host eIF5A (in red) and the mab1C12 antibody recognizing the schizont surface (in green). This new figure also clearly shows release of eIF5A from the schizont surface upon treatment with the MC2646 compound.

Fig 1b. Referring to “parasite nuclei” per host cell rather than “parasites” per cell would be more accurate. The schizont is thought to be a multinuclear syncytium, rather than distinct parasites, so I find “parasite number” misleading.

We agree that ‘parasite nuclei’ may be less ambiguous. We have now modified this in Figure 1a, Figure 1b, Figure 2e, Figure 3a, Figure 4e and Figure 4h. We have modified the text and the figure legends accordingly. We have also modified the labels in Supplementary Figures.

Fig 1f – the colony forming assay shown here is not very clear. Perhaps this is because I only have a low resolution version of the figures.

We have included a high-resolution copy of all the figures for the review process.

Fig 2a. Due to the low resolution of reviewer files I cannot assess how convincing the TEM data is. I would like to see the full resolution figures before commenting on the autophagosome structures induced upon treatment with MC2646. More clear labelling of the parasite would be helpful. And where are the TEM for BL3+ MC2646?

We have included the high-resolution figures for the review process. We have modified Figure 2a to include improved TEM images for BL3 control and BL3 treated with MC2646. We have also added labels to highlight the autophagosome and the annulate lamellae structures.

Figure 2a,b,c,d. You show that MC2646 affects host autophagy in BL3 cells (increase in autophagy related gene pathways and an increase in lipidated LC3). Please also show LC3B puncta formation in BL3 cells before and after treatment (and see previous comment about TEM).

We have added new TEM images to Figure 2a that clearly indicate the autophagosome formation upon treatment with MC2646. To address the reviewer's comments we have also included new data in Supplementary Figure 3. We had added microscopy images and quantification to highlight LC3 puncta formation upon treatment of BL3 cells with the MC2646 compound [See Supplementary Figure 3a,c].

How do you explain that rapamycin and torin1 don't reduce parasite number, if they indeed induce autophagy in infected cells? Did you verify that autophagy is induced upon treatment with these classic autophagy activators? Please expand on this.

Indeed, the autophagy processes are complicated and we have attempted to clarify our findings with additional experiments. We have performed new experiments and added new data to address this issue raised by the reviewer. The Rapamycin and Torin1 compounds induce autophagy in normal cells by inhibiting the mTOR pathway. We added additional data to demonstrate that treatment with Torin1 or Rapamycin did indeed induce autophagy in control BL3 cells, as demonstrated by the formation of LC3 puncta in the presence of BafA1 [new Supplementary Figure 2b]. However, in parasitized cells it would appear that this pathway is bypassed in some way. In contrast, in TBL3 cells neither treatment lead to an increase in puncta or a decrease in parasite nuclei number, compared to controls with BafA1 alone [new Supplementary Figure 2c-d]. Furthermore, treatment with the AMPK inducer 991 significantly reduced parasite nuclei number per host cell in infected cells (although less than Buparvaquone or MC2646) [Figure 3a and Supplementary Figure 2f]. The effects of 991 or MC2646 on parasite survival were rescued by treatment with the AMPK/ULK inhibitor, SBI-0206965 [Supplementary Figure 2f]. These experiments emphasize that MC2646-induced autophagy is via the AMPK pathway. We have modified the text on page page 7 to clarify this issue.

Figure 2c: western blotting with LC3-I and LC3-II. Can you quantify the relative amounts of LC3-II as you do in figure 3f. This might make the data clearer. I do find the accumulation of LC3-II in MC2646 treated cells compared to the control believable (fig 2c, both BL3 and TBL3), although quantification would help. However I'm a bit lost with figure 3e. Why do you state that MC2646 induces autophagy but buparvaquone doesn't? The ratio of LC3-II:I are the same in both treatments. Please explain.

We have quantitated the LC3-II:LC3-I ratio for Figure 2c and highlighted this in the text on page 6. This demonstrates that the ratio is lower in TBL3 cells (0.55) compared to control BL3 cells (0.87) and that LC3-II accumulates in both cells when treated with MC2646 (0.98 and 1.56 respectively). The explanation for Figure 3e is that the comparison between MC2646 and Buparvaquone treatment should be seen in the presence of BafA1 where the ratio of LC3-II:LC3-1 is clearly increased (1.4 for MC2646

compared to 0.9 for Buparvaquone) and the number of LC3 puncta is clearly visible [Figure 3d]. We have clarified this in the text on page 7.

You show nicely that TFEB nuclear translocation is induced by MC2646 treatment. Did you consider knock out, knock down or inhibition of TFEB? Wouldn't you expect this to block the action of MC2646?

We thank the reviewer for this suggestion. We have tried many times to knock-down TFEB (with four different siRNA sequences) without success. The knockdown of TFEB protein was not sufficient for us to confidently conclude, so we have not included these experiments.

Figure 3f. Please provide a quantification of red/yellow puncta in TBL3 cells following treatment with MC2646 +/- Baf1 to support the data shown in fig 3f. The data shown in the IFA images, particularly the bottom panel with Baf1, don't seem to support the description in the text – I don't see many red puncta in the bottom panel, and in the 3rd panel they are at first glance overwhelmed with the large yellow puncta. Maybe it's worth highlighting the red dots with arrows, or a line in the text. Further, it would be interesting to point out the parasite in these cells. Is LC3 excluded from the nucleus following MC2646 treatment? What does this signify? I find that the grey DAPI signal detracts from the red/green data.

This experiment was technically very challenging, as the TBL3 infected cells are very resistant to transfection. For this reason, it was difficult to reproducibly obtain enough transfected cells for satisfactory quantification. That is why we added the Figure 3g panel to quantify convincingly in another cellular model (U2OS). The objective of these two experiments was to demonstrate that MC2646 does indeed induce autophagy in different cell types. We had chosen nonetheless to maintain the Figure 3f rather than remove it altogether. In response to the reviewer's comments, we decided to swap the location of these experiments. We have now move the U2OS experiments to Figure 3f, alongside the quantification in Figure 3g, and we moved the TBL3 experiment to Supplementary Figure 3e. We have added arrows, as requested, to highlight the red dots upon MC2646 treatment. These combined experiments support our conclusion concerning the autophagy-inducing capacity of the MC2646 drug. The manuscript text has been modified accordingly on page 8.

Major point: Figure 4a/b: antibody staining of the schizont surface (TaSP or 1C12) is necessary to show the localization of the drug target as well as eIF5a in infected cells. Why not show a nice 3D reconstruction with 1C12, as you do with mAB414 in figure S3? In figure 4b, the eIF5a staining looks like it overlaps with parasite nuclei rather than the surface of the parasite. How can you exclude that your antibody is cross reacting with Theileria eIF5a?

As stated above, to respond to reviewer's comments we have added a 3D reconstruction of confocal images as the new Figure 4d to show the co-staining of eIF5a and the parasite schizont surface, using an antibody recognizing host eIF5A (in red) and the mab1C12 antibody recognizing the schizont surface (in green). This new figure also clearly shows release of eIF5A from the schizont surface upon treatment with the MC2646 compound. The confocal images suggest that the antibody is detecting a host protein that is recruited to the schizont surface and is then released upon drug treatment. If the antibody detected parasite eIF5A one would expect the staining to remain even after drug treatment. Though without the ability to genetically delete the parasite gene, we cannot completely rule out the possibility of cross-reaction. We have also added a control 'click' compound which is the related Tranylcyproline (TCP) compound which clearly does not localize to the parasite, neither before nor after treatment with MC2646 [Figure 4b]. The text of the manuscript has been modified accordingly on page 8.

Using Click chemistry, you show that eIF5a might be a potential target of MC2646. How do you explain the lack of co-localisation? It seems more likely to me that eIF5a is an indirect target of MC2646 - please expand on this.

The reviewer raises an interesting point. We have tried several biochemical approaches to explore this further, but were unable to demonstrate direct interaction between the drug and the eIF5A protein. Future studies will be directed at define this functional link in more detail. If the effect is indirect, it raises the question of what is the direct target of the MC2646 compound. We hope to get insights into this issue in the future. We have added a note in the text to include the possibility of indirect effects.

Line 191-192. You don't show that reduced eIF5a hypusination causes a decrease in TEFB and ATG3, you just show that these two observations correlate with each other. Please modify text. You don't discuss Atg3 expression following MC2646 treatment, with or without GC7. If anything, ATG3 seems to increase following MC2646 treatment, regardless of treatment with GC7. The expression of p62 is not discussed at all and does not seem to be impacted by MC2646 treatment. Have I missed something?

We thank the reviewer for this careful reading of the manuscript. Indeed, we had expected to see a direct effect on Atg3 and/or p62 protein levels. This is especially true as the bovine Atg3 protein contains a 'PPPP' polyproline motif and the p62/SQSTM1 protein has a 'PPP' motif, which are characteristic of eIF5A functional targets. We have added a comment in the discussion to mention this point.

Reviewer #2

In this manuscript, Villares and colleagues discovered a novel compound named MC2646 that can effectively suppress the growth of Theileria parasites in host cells. Mechanistically, Theileria parasites suppressed host cell autophagy to facilitate their survival, and MC2646 efficiently induced autophagy probably via activating AMPK to suppress parasites growth. Indeed, MC2646 can be a very useful tool for the autophagy field. The authors also discovered that eIF5A may be sequestered by intracellular parasites, although the physiological impact of such sequestration remains unclear. Pharmaceutical inhibition of eIF5A inhibited autophagy and abrogated MC2646's parasites-inhibition effect.

We thank Reviewer 2 for careful reading of our manuscript and for insightful questions.

The following points may help further improve the manuscript:

1. One or two paragraphs of introduction would be very helpful for readers to learn the background knowledge.

We thank the reviewer for inviting us to add more introductory text. We were initially limited by the journal style and limits. We have added text to the introduction to provide more context and background.

2. Line 51 and Fig1a. Although it's published, it is still helpful to show some examples of the screen images of positive and negative controls, to directly show readers how each parameter is measured.

Since our submission, our initial study describing the screening conditions has been published in *Communications Biology* and we have added this to the reference list [Reference #25]. However, to address the point raised by the Reviewer we have added a new Supplementary Figure 1a which shows examples of the microscopy images from the screen and highlights the loss of H3K18me1 labelling upon treatment with the MC2646 compound. We modified the text accordingly [page 4, line 106].

3. Line 65 and Fig 1c: Apoptosis = “sub G1 population”? How this assay of cell cycle and apoptosis was performed? Original images could be shown as an example (in supplementary figures), and more details can be provided in the figure legend.

To reply to the Reviewer's request, we have added images in the new Supplementary Figure 1b to show flow cytometry profiles of BL3 and TBL3 cells treated with Buparvaquone or MC2646 drugs. These are examples of the experiments used to generate the data presented in Figure 1c.

4. Fig 2d. Do parasites localize within autophagosomes when treated with MC2646? Could these parasites be digested/killed by lysosomes?

This is a really good question and one that we struggled with. Although, we sometimes observed images that suggested this was the case, we have no firm evidence to categorically state this.

5. Fig 2e. MC2646 still reduced parasites load in the presence of BafA1 (comparing the last two columns). Does that suggest that MC2646 may at least partially work in an autophagy/lysosome-independent manner to suppress parasites survival?

This is an interesting question. Unfortunately, we do not have clear evidence to definitively address this issue. We must emphasize that one confounding factor is the timing of the different drug incubations. For example, we treated cells for 24 hours with the MC2646, before adding BafA1 for the last 2-3 hours. This means that the effects of MC2646 on parasite survival may already be advanced before the addition of BafA1. We have added a note in the text to address this issue [page 6]. The important message of the experiment presented in Figure 2e was to show that EBSS treatment was as effective as MC2646 in reducing parasite load through the induction of autophagy and that this was primarily reversed upon BafA1 treatment.

6. Fig 3a. Do all of these tested drugs work efficiently to induce or inhibit autophagy as expected? It is confusing why 991 but not Rap or Torin1 can significantly suppress parasites, as rap and torin1 should be potent autophagy inducers as well.

As explained above in response to Reviewer 1, we have added new data to show that the mTOR inhibitors Rapamycin and Torin1 induced autophagy in control BL3 cells, but not in TBL3 cells [Supplementary Figure 2b]. The results with 991 treatment suggest that the autophagic effect is probably independent of mTOR, but is AMPK dependent. We have added to the text to clarify this point.

7. Fig 3f. Quantification is required. Also, BafA1 treatment did not increase the number of green/yellow dots (autophagosomes), nor reducing the number of red dots (functional lysosomes) in control cells. Does BafA1 work well in TBL3 cells?

As explained in our response above to Reviewer #1, the technical challenges make it impossible to obtain enough transfected cells for satisfactory quantification. As explained above, we have now swapped the experiments with those in U2OS cells [Figure 3f-g]. Furthermore, the new experiments and quantification presented in Supplementary Figure 3b-e show clearly that treatment with BafA1 can induce clear LC3 puncta in TBL3 cells.

8. Fig 3g. Again, red dots are active lysosomes. How to explain that BafA1 treatment did not cause reduction of red dots? The autophagy induction by MC2646 is pretty convincing though.

The data presented in Figure 3g and the microscopy images presented in Supplementary Figure 3e demonstrate that that MC2646 clearly induces autophagosomes and that the red dots are reduced upon treatment with BafA1. This experiment is difficult to do in TBL3 cells because of the poor transfection efficiency and technical difficulties of quantification. To clarify this point we have included the U2OS data in the Figure 3f and moved the TBL3 data to Supplementary Figure 3.

9. Fig 4a. A negative control would be helpful here by adding another alkyne compound, or MC2646, to cells for Click to confirm that the staining is specific. Also, it seems that with the alkyne modification of MC2646 abrogated its inhibitory function because it does not suppress parasite growth in "Ctrl" cells? If that's the case, the interactome of MC2646 could be changed by the modification as well.

We thank the reviewer for these comments. To address this, we have added new data to Figure 4b. Here, we show a control compound which is a 'click' form of the TCP molecule from which the MC2646 compound was derived. Here, it is clear that the TCP control does not give the dot structures, on the parasite schizont surface in TBL3 cells, that are characteristic of the 'click' MC2646 compound. Furthermore, treatment with MC2646 reduced the schizont size, but the TCP-click molecule remained diffuse throughout the cell. We have added a comment in the text of the manuscript about this chemical control compound [page 8]. We also need to clarify the experiments presented in Figure 4a. The click compound (named MC4404) used in this experiment was used simply as a probe to mark the targeting of the MC2646 drug to the parasite surface. As the MC4404 compound is toxic, TBL3 cells were incubated with or without the MC2646 autophagy inducer and MC4404 was just added for 30 minutes for the visualization. We have added a note in the text to clarify this [page 8].

10. Fig 4b. eIF5A localizes to ribosomes to facilitate translation, thus shouldn't form large puncta as its functioning form. Could the authors confirm the staining specificity of eIF5A in this figure? It might be hard to directly image ribosomes, but would be helpful if the authors could confirm the finding using biochemical approaches to show that eIF5A is sequestered by parasites, and released to the cytosol upon MC2646 treatment.

11. Fig 4. Generally, this figure shows two very interesting observations: 1) the potential interaction between eIF5A and parasites, and 2) GC7 treatment almost fully blocked MC2646's parasite-suppressing effect. Although MC2646 may not directly interact with eIF5A, several interesting questions still arise: How does the infection alter host translation, including the general translation rate (OPP-Click assay), or the translome in depth (ribosome pull-down sequencing/polysome profiling/ribosomal profiling)? eIF5A may specifically promote translation of hard-to-read regions such as polyprolines. Does the parasite infection lead to translation stalls at those regions?

Here the reviewer raises some fascinating points. As discussed above, despite several different approaches, we have not been able to demonstrate whether eIF5A interacts directly with the MC2646 drug or how the sequestration and release of eIF5A impact the translome of the host cell. While

these are important points, we have been unable to come up with convincing data at this point and feel that the general question of how the parasite impacts the transcriptome of the host cell and how the induction of autophagy impacts ribosome function is beyond the scope of this study. We hope to be able to establish ribosome profiling analysis in the future, but we consider that this would be a whole new research project in itself.

12. What's the relationship between AMPK and eIF5A? Would GC7 treatment also abolish 991's effects on activating AMPK and suppressing parasites growth?

This is an interesting point. The relationship between the AMPK pathway and eIF5A has not been explored in the literature. We have added new data to show that treatment combining 991 and GC7 appears to largely rescue parasite survival [Supplementary Figure 5c]. This suggests that eIF5A hypusination is downstream of AMPK. We have modified the text to mention these data [page 9].

Minor points:

13. Line 32: It is "unknown"...

We have modified the text.

14. Line 51: histone "marker"

We have modified the text.

15. Fig 1a. The control lines are not red as indicated by the legend.

We have modified the text.

16. It seems that eIF5A is not the major finding in this paper because eIF5A only came up in Fig 4 without a clear mechanism being revealed, thus the short title "Theileria sequesters eIF5a" is not very accurate. Similarly, the main title shall also be adjusted and focus more on the novel autophagy-inducing compound.

We have changed the short title to 'Theileria escapes elimination by autophagy'

Additional modifications

Our study of a drug screen for anti-*Theileria* compounds has now been published in *Communications Biology* (initially listed on page 3, line 49 as (*Communications Biology*, in press)). This has now been added to the list of references (page 4, line 104, Reference 25).

In reformatting the manuscript for the **Nature Communications** style, we have removed the references from the Abstract and added an Introduction section.

We have added more details of the EM methodology on page 23.

All the modifications to the text are all indicated in blue.

REVIEWERS' COMMENTS

Reviewer #1 (Remarks to the Author):

I appreciate all the changes the authors have made, and find the revised manuscript much clearer. It's a very exciting story! The authors have addressed all my comments adequately. I do have a few minor comments which could help with the clarity of the manuscript:

Line 90 «induces AN (not and) AMPK-dependent pathway»

Line 115: "...We validated the screen results, testing the MC2646 compound (1 μ M for 48 hours) on infected macrophages (Tac12 cells) OR B lymphocytes infected"

Line 142 / figure 1g/1h: it looks as though mmp9 is shown to be downregulated in TBL3 vs BL3 (blue in inner track figure 1g), but then up-regulated following buparvaquone treatment (red in outer track fig 1g). In fig 1h it looks as though mmp9 is only very slightly downregulated following bup treatment (pale blue, top track fig 1h), while following MC2646 it is significantly downregulated – in the text you say modest gene regulation after MC2646 treatment. Are the legends / colour codes mixed up, or have I misunderstood something? Please double check this and adjust the text /legend accordingly to make things clearer.

Line 171: «...The observation that BafA1 blocks the EBSS-induced parasite loss more than MC2646 [Figure 2e], suggests that MC2646 may also induce other pathways.....» I find this rather unclear, please re-write. On first reading, I thought you expected addition of MC2646 to rescue the EBSS-induced parasite loss which of course makes no sense. How about changing to «The observation that BafA1 blocks the EBSS-induced parasite loss more than it does MC2646-induced parasite loss"»

Fig 2e: I would have thought it would be more meaningful to do an experiment in complete media (not EBSS) with MC2646 + BafA1? This appears to be missing. You seem to have done this experiment (fig 3d) for quantification of LC3 puncta, but not correlated it with parasite survival.

Figure 2d/2e: please include a figure to show that starvation does indeed induce autophagy in these cells. You show that LC3 puncta are barely detectable in untreated TBL3. These are induced by MC2646. But you do not show that autophagy can be induced by the classical approach of starvation, which does reduce parasite nuclei. I find this is important, because you next state that rapamycin (mTor inhibition) does not induce autophagy. Please comment on the difference between starvation (which has the effect of inhibiting mTor?) and mTor chemical inhibition to induce autophagy.

Line 198: «...As MC2646 treatment appears to reinitiate autophagy and activate the AMPK pathway....» this statement is only based on one western blot with anti p-AMPK. Is the activation of the AMPK pathway also supported by your RNAseq data? A sentence about this would be helpful.

Line 207 : i don't know why you talk about p62 here? Not very clear sentence.

Figure 4A: it would make it clearer to label the Click panel of panel a «MC4404 + Click» to be consistent with fig 4B, and to make the figure instantly understandable.

Line 316: eIF5a-reGulated

Reviewer #2 (Remarks to the Author):

The authors have made a diligent effort to respond to my concerns. Although it remains confusing why rapamycin/torin 1 don't suppress parasites even when they could still efficiently induce autophagy, what the direct target of MC2646 is, and how eIF5A is sequestered, the major conclusions listed in the abstract are convincingly supported. I believe that the discoveries will attract broad interest in the autophagy and the Theileria infection fields.

One minor point: The written format of eIF5A or eIF5a should be consistent throughout the manuscript. Both are seen in literatures, but I would recommend eIF5A, or even EIF5A, because protein names should be capitalised.

REVIEWERS' COMMENTS

All changes in the text are highlighted in red

Reviewer #1 (Remarks to the Author):

I appreciate all the changes the authors have made, and find the revised manuscript much clearer. It's a very exciting story! The authors have addressed all my comments adequately. I do have a few minor comments which could help with the clarity of the manuscript:

We are delighted that the reviewer is satisfied by the revised manuscript.

Line 90 «induces AN (not and) AMPK-dependent pathway»

We have modified the text.

Line 115: "...We validated the screen results, testing the MC2646 compound (1 μ M for 48 hours) on infected macrophages (Tac12 cells) OR B lymphocytes infected"

We have modified the text.

Line 142 / figure 1g/1h: it looks as though mmp9 is shown to be downregulated in TBL3 vs BL3 (blue in inner track figure 1g), but then up-regulated following buparvaquone treatment (red in outer track fig 1g). In fig 1h it looks as though mmp9 is only very slightly downregulated following bup treatment (pale blue, top track fig 1h), while following MC2646 it is significantly downregulated – in the text you say modest gene regulation after MC2646 treatment. Are the legends / colour codes mixed up, or have I misunderstood something? Please double check this and adjust the text /legend accordingly to make things clearer.

Indeed, in the re-submitted version of the manuscript the MC2646 and Bup labels were accidentally reversed in Figure 1h. This has been corrected.

Line 171: «....The observation that BafA1 blocks the EBSS-induced parasite loss more than MC2646 [Figure 2e], suggests that MC2646 may also induce other pathways.....» I find this rather unclear, please re-write. On first reading, I thought you expected addition of MC2646 to rescue the EBSS-induced parasite loss which of course makes no sense. How about changing to «The observation that BafA1 blocks the EBSS-induced parasite loss more than it does MC2646-induced parasite loss"»

We have modified the text.

Fig 2e: I would have thought it would be more meaningful to do an experiment in complete media (not EBSS) with MC2646 + BafA1? This appears to be missing. You seem to have done this experiment (fig 3d) for quantification of LC3 puncta, but not correlated it with parasite survival.

In Supplementary Figure 3d, we show in parallel the effect on autophagosomes (LC3-II and p62-positive structures) as well as parasite survival (parasite nuclei per host cell in Ctrl or MC2646 treatment +/- BafA1).

Figure 2d/2e: please include a figure to show that starvation does indeed induce autophagy in these cells. You show that LC3 puncta are barely detectable in untreated TBL3. These are induced by MC2646. But you do not show that autophagy can be induced by the classical approach of starvation, which does reduce parasite nuclei. I find this is important, because you next state that rapamycin (mTor inhibition) does not induce autophagy. Please comment on the difference between starvation (which has the effect of inhibiting mTor?) and mTor chemical inhibition to induce autophagy.

We have added a new panel Figure 2f that shows the quantification of LC3 puncta in the EBSS conditions.

We know that the Torin1 treatment works because it increases autophagic flux in non-infected cells, but its effect is not enough to restore the autophagic flux inhibited by the parasite infection. Starvation with EBSS media indeed inhibits mTor, but seems to have a more global effect (even on the parasite) than the specific mTor chemical inhibition.

Line 198: «...As MC2646 treatment appears to reinitiate autophagy and activate the AMPK pathway...» this statement is only based on one western blot with anti p-AMPK. Is the activation of the AMPK pathway also supported by your RNAseq data? A sentence about this would be helpful.

Our RNAseq data support the AMPK activation as we observed an upregulation of several genes associated with AMPK activation. These include *ULK1*, and consequently *ULK1* targets like *ATG14* or *Beclin1*. These changes are, however, relatively modest and not highly statistically significant.

Line 207 : i don't know why you talk about p62 here? Not very clear sentence.

We have modified this sentence for simplicity and clarification.

Figure 4A: it would make it clearer to label the Click panel of panel a «MC4404 + Click» to be consistent with fig 4B, and to make the figure instantly understandable.

We have modified the Figure 4 with the label 'MC4404+Click' as suggested.

Line 316: eIF5a-reGulated

We have modified the text.

Reviewer #2 (Remarks to the Author):

The authors have made a diligent effort to respond to my concerns. Although it remains confusing why rapamycin/torin 1 don't suppress parasites even when they could still efficiently induce autophagy, what the direct target of MC2646 is, and how eIF5A is sequestered, the major conclusions listed in the abstract are convincingly supported. I believe that the discoveries will attract broad interest in the autophagy and the Theileria infection fields.

We thank the reviewer for their enthusiasm.

One minor point: The written format of eIF5A or eIF5a should be consistent throughout the manuscript. Both are seen in literatures, but I would recommend eIF5A, or even EIF5A, because protein names should be capitalised.

We apologize for this inconsistency. We have modified the protein to eIF5A throughout the text.